# Fungal Grapevine Trunk Diseases in Romanian Vineyards in the Context of the International Situation

**DOI:** 10.3390/pathogens11091006

**Published:** 2022-09-02

**Authors:** Maria-Doinița Muntean, Ana-Maria Drăgulinescu, Liliana Lucia Tomoiagă, Maria Comșa, Horia-Silviu Răcoare, Alexandra Doina Sîrbu, Veronica Sanda Chedea

**Affiliations:** 1Research Station for Viticulture and Enology Blaj (SCDVV Blaj), 515400 Blaj, Romania; 2Electronics, Telecommunication and Information Technology Faculty, University Politehnica of Bucharest (UPB), 060042 Bucharest, Romania

**Keywords:** fungal pathogens, grapevine trunk diseases (GTDs), Romanian vineyards, GTD management

## Abstract

*Vitis vinifera*, known as the common grape vine, represents one of the most important fruit crops in the world. Romania is a wine-producing country with a rich and long tradition in viticulture. In the last decade, increasing reports of damage caused by grapevine trunk diseases (GTDs) have raised concerns in all wine producing countries. Up to now, no study was performed regarding the GTDs situation in Romania, an important grapevine grower in Europe. In this study, we aim, after a comprehensive presentation of the fungal GTDs worldwide, to review the scientific information related to these diseases in Romania in order to open a national platform in an international framework. In order to achieve this, we consulted over 500 references from different scientific databases and cited 309 of them. Our review concludes that, in Romania, there is little amount of available literature on this matter. Three out of six fungal GTDs are reported and well documented in all of the Romanian viticultural zones (except for viticultural zone 4). These are Eutypa dieback, Phomopsis dieback, and Esca disease. Of the fungal pathogens considered responsible *Eutypa lata*, *Phomopsis viticola* and *Stereum hirsutum* are the most studied and well documented in Romania. Management measures are quite limited, and they mostly include preventive measures to stop the GTDs spread and the removal of affected grapevines.

## 1. Introduction

*Vitis vinifera* L., known as the common grapevine, represents one of the most important fruit crops in the world, having also great economic importance due to the high commercial value of the fresh table grapes, dried fruits, and wine productions. Vineyards of *V. vinifera* are mainly located in the Mediterranean region, Central Europe, and southwestern Asia, from Morocco and Portugal to north to southern Germany and east to northern Iran [1].

Romania, located between 43°37′–48°15′ N lat and 20°15′–29°44′ E long, in Eastern Europe, having a temperate continental climate, Dfb and Dfa in a Koppen–Geiger climate updated classification [2,3], is a wine-producing country with a rich and long tradition in viticulture. If before 1989, centralized cultivation technologies were applied, after 1989, there was a change in the field, mainly due to the fragmentation of the viticultural surfaces and the increase of the cultivated surfaces with direct producing hybrids, to the detriment of the noble varieties with tradition [4]. This situation caused a regression of Romanian viticulture. Because the sector is one of major importance for the country’s economy, the Romanian Government through the Minister of Agriculture and Rural Development has developed a series of programs to support and boost it. Therefore, restructuring or reconversion of vineyards had the greatest impact on the grapevine sector, with Romanian producers managing to restructure or modernize areas with vines that correspond to current market requirements [5].

In 2020, the global grapevine cultivation area reached 6,950,930 ha, with a total production of 78,034,332 tones. Romania is ranked in 10th place with 175,590 ha of harvested area and 932,770 tons of grape production [6] (Figure 1).

As shown in Figure 2, the European continent leads in terms of *Vitis vinifera* harvested areas, with 3,457,119 ha representing almost 50% (49.74%) of world areas [6]. In the European ranking, Romania occupies the 5th place both on the areas harvested and production criteria, being outranked by Portugal, Italy, France, and Spain [6].

Grapevine can be affected by many different biotic stress agents, including pathogenic fungi that induce severe disease symptoms on different plant organs [7]. The diseases affecting the woody tissues (trunk and cordons) are the so-called grapevine trunk diseases (GTDs), and they are often the most important and most destructive grapevine diseases in all of the areas where grapevines are cultivated [7,8] causing considerable yield loss and reducing vineyard lifespans and thus generating substantial economic losses in the wine industry [9,10,11,12,13,14,15,16,17,18,19].

GTDs are a handful of complex diseases caused by many taxonomically unrelated fungi [20,21]. These pathogens cause different vascular and foliar symptoms, which cause overall decline and eventual death of grapevines. [1,11,22]. Caused by xylem-colonizing fungi, GTDs are slow-progression diseases, whose symptoms sometimes take several years to appear after infection [19]. Moreover, GTDs are known to have an inconsistent expression of the symptoms from year to year on individual grapevines [11,19,23,24], with abiotic and biotic stresses being likely to play important roles in GTDs symptoms development [1,19,24,25,26]. In addition, there are several reports that reveal GTDs fungi isolated from asymptomatic tissues, meaning that these fungi may act as latent pathogens [1,19,25]. The symptoms of a diseased plant are typically the result of a complex interaction between different factors. The most important of these factors are the secondary metabolites with phytotoxic activity produced by the fungi involved in the GTDs [7], which are translocated to the leaves via the xylem and are suspected of being involved in the expression of foliar symptoms [7,27]. A wide range of pathogen derived biochemicals are currently studied to understand their role in GTD foliar symptom expression, such as secreted proteins [26,28,29], exopolysaccharides [7,26,30], and secondary metabolites [7,26,29,31]. Comprehensive reviews reporting the biochemical characterization of these compounds and their role in the induction of GTDs are reported by Masi et al. [7] and Schilling et al. [29], where the main classes of metabolites and their phytotoxins (Naphthalenones; Isocoumarins; Cyclohexene Epoxides and Related Compounds; Phenols; Chromanones and Related Phytotoxins; Jasmonic Acid and Its Esters; Dihydrofuranones and Related Phytotoxins; Miscellanea; Exopolysaccharides; Polypeptides) are described. All these factors make GTDs identification and management quite difficult [1].

Increasing reports of damage caused by GTDs in the last 20 years have rapidly raised concerns in all wine producing countries [22,32]. OIV provides some data from different countries about the impact of these diseases: for example, in Spain since 2003 (year of the sodium arsenate prohibition in Spain), GTDs have grown from 1.8% vineyards degree affections to 10.5% in 2007 [32,33]. Moreover, close to 13% of French vineyard were affected by trunk diseases according to the survey led by the DGAL in 2012 [32,34]. GTDs are also a growing problem for many vineyards in all regions of Italy where epidemiological studies showed 60% to 80% GTDs incidence reached in some old vineyards [32,35]. Other similar reports from Portugal, Argentina, Turkey, USA, Australia, New Zealand are presented by OIV in a review on Grapevine Trunk Diseases from 2016.

Knowledge of the distribution of GTDs and the main factors associated with their development is essential to predict the GTDs spread and to improve disease management [1,21,22] in order to reduce their negative economic impact. Up to now, no study was performed in order to show the situation of GTDs in Romania, an important grapevine grower in Europe and in the world. We aim in this paper to give a comprehensive presentation of the fungal GTDs worldwide and to review the scientific information related to these diseases in Romania in order to open a national platform in an international framework for studying this matter.

## 2. State of the Art in GTDs Understanding at Global Level

Among all causes of grapevine decline, GTDs are major concerns for grape growers all around the world [36], and they are primarily caused by taxonomically unrelated Ascomycete fungi and to lesser extent by several Basidiomycetes [1,19]. The major GTDs gather diseases associated with either one particular fungal species, for example, Eutypa Dieback, or with fungal species complexes, for example, Esca [29].

In the scientific literature, there are some studies regarding the action mechanism of the pathogenic fungi responsible for GTDs [29,36,37,38,39,40,41]. It is generally accepted that pathogenic fungi (that are soilborne, airborne, or both) use vine wounds (natural, mechanical, or pruning wounds) as a point of entry to the plant vascular system, colonize the host and decay the wood, causing an irreversible loss of function of the xylem and phloem elements that results in dieback and wilt symptoms [17,38,42,43,44,45,46]. The life cycle and epidemiology are very similar for all the known fungi involved in grapevine trunk diseases [11,32]. These diseases are cryptic, and their symptoms usually take several years to develop. As such, they are insidious and difficult to observe [32].

Grapevines have the highest risk of infection during the pruning period, from late fall to early spring, because of the high number of wounds made on a single grapevine and the frequency of rain events that occur during that period. Grapevine wounds remain susceptible to infection by these fungi for several weeks [44,45,47,48]. Plugging of the xylem and phloem elements and decay of the wood follow infection, impairing translocation of water and nutrients and leading to the decline of the grapevine [38].

The term Grapevine Trunk Diseases accounts for a large number of diseases such as Esca disease, Eutypa dieback, Phomopsis dieback (excoriosis), Botryosphaeria dieback, Petri disease, and Blackfoot disease [1,11,21,24,48]. Overall, Esca, Eutypa dieback, Botryosphaeria dieback, and Phomopsis dieback (excoriosis) are the leading GTDs encountered mostly in old vineyards presenting visible symptoms, while Petri disease and Blackfoot disease are major diseases mostly found in young vineyards, and they present less evident symptoms, somehow like those of the early stages of GTDs [24,27,48,49].

Environmental factors and host stress (abiotic factors) such as malnutrition, poor drainage, soil compaction, heavy crop loads on young plants, planting of vines in poorly prepared soil, and improper plant holes play an important role in the development of Blackfoot and Petri diseases [40]. As time passes, the biotic factors aggravate the grapevine health so the symptoms of GTDs intensify [27,36,50,51,52,53].

Grapevines can be affected by one or more GTDs at the same time, as individual plants can be infected by different pathogens, due to co-occurrence of multiple infections throughout a season, and over years. This produces overlapping of the symptoms, which makes their association with the specific responsible fungi particularly difficult to define and detection of causal pathogens challenging [1,16,17,29]. The aggressiveness and symptoms caused by fungal pathogens associated with GTDs differ significantly between grapevine-growing regions and vary depending on cultivars [48,54]. Moreover, GTD symptoms are known to be expressed inconsistently from year to year on individual grapevines [11,24,55].

### 2.1. Petri Disease

Petri disease is a vascular disease of grapevines associated with decline and dieback of young grapevines that has caused significant losses in newly planted vineyards and has been increasingly reported in wine-growing areas worldwide [1,40,56,57,58,59,60,61,62,63,64,65].

#### 2.1.1. Fungi Involved

The main causal agents of Petri disease are the pathogenic fungi *Phaeomoniella chlamydospora* and numerous species of *Phaeoacremonium,* of which notably are *P. aleophilum* [58,61,62,66,67] and *P. inflatipes* [27,68]. Pycnidial synanamorph of *Phaeomoniella chlamydospora* has been observed on grapevines in the field [68,69] and conidia of *P. aleophilum* and *P. inflatipes* were trapped or detected in vineyards [61,68,70,71,72,73].

A major mean of spread is believed to be via infected propagation material, specifically rootstock material [27,56,61,68,74]. *Phaeomoniella chlamydospora* DNA was detected by nested-PCR in hydration and drench water, grafting tools, callusing media, and soils in New Zealand and South African grapevine nurseries [60,62,75,76,77,78,79,80] indicating that they are potential inoculum sources for this pathogen. Fourie and Halleen [66] isolated *Phaeomoniella chlamydospora* from the stained wood of rooted cuttings and grafted grapevines that had failed, indicating that contamination with untreated water, soil or dust occurred during the propagation process.

Infected soil has also been considered as a potential inoculum source since *Phaeomoniella chlamydospora* was found in nursery and vineyard soil by means of conventional species-specific PCR [77] and nested-PCR [75,81]. Rooney et al. [82] and Whiteman et al. [75] also detected this pathogen from soil surrounding an infected rootstock mother vine [62,68]. *Phaeomoniella chlamydospora* and *Phaeoacremonium* spp. can inhibit graft healing by grapevine rootstock and scion cultivars [56,62,83].

#### 2.1.2. Symptoms

Formerly known as black goo decline, brown wood streaking, and young vine decline of grapevine [27,66], Petri disease affects mostly 1- to 5-year-old grapevines [66]. External symptoms include stunted growth, interveinal chlorosis, leaf necrosis, decline, and possibly dieback [61,66]. Dissected vines show a typical black discoloration of the xylem vessels, which is a result of tyloses, gums, and phenolic compounds formed inside these vessels by the host in response to the fungus growing in and around the xylem vessels [27,61,66].

It is believed that the host is predisposed to the pathogenic phase of this fungus by stress, in particular water stress [66,84]. Blocked xylem vessels accentuate the water stress and lead to insufficient water and nutrient supply to the vegetative plant parts. This consequently leads to symptom expression, which usually occurs during periods of high-water demand [27,62,66]. Furthermore, the pathogens produce phytotoxic metabolites that contribute to symptom expression [29,66,85].

#### 2.1.3. Management

In recent years, special attention has been given to the development of procedures and products to prevent or reduce Petri disease infection of the woody tissues of grapevine propagation material. Biological and physical treatments to control Petri disease such as hot-water treatments were studied by Fourie and Halleen [68], Waite and Morton [86], and Gramaje et al. [62,67], but to date, no chemical products have been registered for this specific use, despite promising results both in vitro and in vivo [62,67].

### 2.2. Blackfoot Disease

Grapevine Blackfoot disease is well-documented, referred to as one of the most destructive GTD, especially in nurseries and young vineyards in various countries worldwide [32,66,87]. It was first reported in 1961 in France and afterwards in many grapevines growing regions, such as Australia [88,89], California [90,91,92], Chile [93,94], Italy [95], Lebanon [96], New Zealand [57], Portugal [97,98,99], Spain [100,101,102,103,104], South Africa, Uruguay [105], Tasmania [106], Sicily [107], Pennsylvania, USA [108], Greece [109], South Africa [110,111], British Columbia [112], Turkey [113,114], Iran [115], Peru, Switzerland, Japan, Brazil, Canada, Czech Republic, and Bulgaria [87].

#### 2.2.1. Symptoms

Blackfoot disease primarily affects young grapevines of up to 8 years of age [87]. Characteristic symptoms of blackfoot disease include a reduction in root biomass and root hairs with sunken and necrotic root lesions [98,101,105,116,117,118]. In some cases, the rootstock diameter of older vines is thinner below the second tier. To compensate for the loss of functional roots, a second crown of horizontally growing roots is sometimes formed close to the soil surface. Removal of rootstock bark reveals black discoloration and necrosis of wood tissue which develops from the base of the rootstock. The pith is also compacted and discolored [111,116,118,119,120].

External symptoms show reduced vigor with small-sized trunks, shortened internodes, and uneven wood maturity [103,118]. Foliar symptoms associated with blackfoot disease are practically indistinguishable from those observed in grapevines affected by Petri disease and include delayed bud-break, chlorotic foliage with necrotic margins, overall stunting (sparse foliage and small leaves), and wilting of leaves or entire shoots [103,118]. This also resembles symptomatology associated with abiotic disorders such as spring frost, winter damage, nutrient deficiency, and/or water stress [1,103].

When young vines are infected, death occurs quickly. Nevertheless, as the vine ages, infection results in a more gradual decline, and death might only occur after a year or even more [103,121]. Disease symptoms on mature vines (5 years and older) are noticed early in the growing season. Affected vines achieve poor new growth, fail to form shoots after winter dormancy, and die by mid-summer. Often shoots also dry and die during the summer. Vines with reduced vegetative growth also die during the subsequent dormant winter period [103,116].

#### 2.2.2. Risk Factors

Disease risk may be increased by the stresses imposed on young grapevines in nurseries and vineyards [118]. Environmental factors and vineyard management practices (poor drainage, soil compaction, and inadequate planting holes, all leading to poor root development), as well as poor nutrition, heavy cropping of young vines, and the effects of pests and pathogens could be considered as stress factors [118,122]. Moreover, high temperatures during summer, play an important role in symptom expression, since the compromised root and vascular system of diseased plants would not be able to supply enough water to compensate for the high transpiration rate [118,120]. The processes of nursery propagation and vineyard establishment include many practices that cause stress on young vines. During the grapevine propagation process, wounds produced during cutting and bench-grafting, the early development of roots and shoots in the nursery field, uprooting and trimming, extended cold storage, and excessive time in containers prior to establishment in the vineyard are all traumatic to the young plants. In addition, after planting out in the field, these vines are again stressed by the need to develop roots and shoots in an environment that is often selected to limit shoot growth [118,122,123].

#### 2.2.3. Fungi Involved

The causal agents associated with blackfoot disease include 7 fungal genera: *Campylocarpon*, *Cylindrocarpon*, *Cylindrocladiella*, *Dactylonectria*, *Ilyonectria*, *Neonectria,* and *Thelonectria* [87,118,124,125]. Many studies state that there are 24 *Cylindrocarpon* species associated with the blackfoot disease of grapevine [39,87,94,98,112,115,117,124] which are mostly soil-borne pathogens [57,116,124,126,127,128] and can infect roots and stem bases [87]. They can survive in soil and act as a source of inoculum for the next disease cycle even if infected plants are removed. Moreover, inocula can spread throughout the field when infested soil is shifted by water or machinery [57,87,116].

*Cylindrocarpon*-like asexual morphs produce conidia, and some species also produce chlamydospores in culture. The conidia are spread in soil water, and the chlamydospores can allow these pathogens to survive in the soil for extended periods of time [39,103]. Infection can occur through the small wounds made when roots break off during the planting process, through the incomplete callusing of the lower trunk, or through wounds made in the grapevine propagation process, such as disbudding wounds, from which the infection progresses downward to the base of the trunk [103,116].

Recent research in Spain has also suggested that blackfoot causing fungi could be latent in visually healthy grapevine nursery stock [103,129]. Much of the current knowledge on blackfoot disease pathogens of grapevine has been derived from research based on populations isolated from vines displaying foliar or internal wood disease symptoms [103]. Blackfoot disease pathogens have been detected, identified, and quantified in soil samples by PCR-based methods [77,129,130,131,132,133] or by dilution plating technique together with the use of a semi-selective medium [103,134].

#### 2.2.4. Management

Presently, there are no curative control measures available to eradicate blackfoot pathogens in nurseries or vineyards [118,135,136]. In recent years, studies have been mainly focused on the development of procedures and chemical products able to prevent or reduce blackfoot disease infection during the propagation process with promising results including, the use of hot water treatments [40,136], biological control (ex. with Trichoderma treatments) [104,111], applications of chitosan [137], use of arbuscular mycorrhizal-AM [138] fungi or fungicides [62,67,139]. Other management strategies identified in literature include host resistance [121], biofumigation [140], incorporating green crops of Brassica species (mustard and rape) into the soil [141], and composting which is also known to suppress pathogenic fungal species [108].

In vineyards, management strategies recommended for prevention and disease management mostly involve the prevention and/or correction of predisposing stress situations [118,136]. Recently there have been advances in the development of procedures and products to prevent or reduce the infection of woody tissue of grapevine: good hygiene and wound protection are of the utmost importance in terms of obtaining healthy vines, which is fundamental to the successful beginning and sustainability of all grape vineyards [40,118]. In this context, in order to improve the quality of grapevine planting material, a sanitation program is needed. Moreover, chemical, physical, and biological control as well as other management strategies should be used to decrease the incidence and severity of blackfoot pathogens during the nursery propagation process and during the growing season in vineyards [118].

### 2.3. Botryosphaeria Dieback

Botryosphaeria dieback is one of the most important, harmful, widespread, and complex GTD worldwide [11,142,143,144] listed among the main biotic threats to the economic sustainability of viticulture, reducing yield and longevity of grapevines [1,104]. Yield losses associated with Botryosphaeria dieback of 30.7% (in 2010) and 42% (in 2018) have been reported in Chilean vineyards [145]. Similar studies estimated yield losses ranging between 25 and 30% in France [146].

Botryosphaeria dieback has been reported since the 1970s [28,147]. In 1974, Lehoczky, first described the disease associated with *Diplodia mutila*, identified in the Tokaj grape-growing region of Hungary, as Black dead arm (BDA). Later, other *Botryosphaeriaceae* species were also associated with the disease, and since a various number of species included in the *Botryosphaeriaceae* family [148] have been isolated from grapevine, Úrbez-Torres [149] and Úrbez-Torres et al. [20] proposed Botryosphaeria dieback as disease name to include all of the symptoms caused by *Botryosphaeriaceae* species on grapevine [11].

Infection is associated with the annual pruning wounds or other physical damage [1]. Conidia are released from pycnidia with the onset of rain and spread by wind or rain splash to infect exposed wound surfaces [142]. The conidia germinate and colonize the woody tissue via xylem vessels damaging the vascular system of the vine thus causing, cankers, dieback, and decline resulting in reduced vegetative growth and yield [142,150,151].

#### 2.3.1. Fungi Involved

Currently, there are 26 botryosphaeriaceaous species in the genera *Botryosphaeria*, *Diplodia*, *Dothiorella*, *Lasiodiplodia*, *Neofusicoccum*, *Neoscytalidium*, *Phaeobotryosphaeria*, and *Spencermartinsia* [1,144,149,152,153,154,155,156] associated with Botryosphaeria dieback. Among them, *Diplodia seriata* is one of the most frequently isolated species from diseased vines in Australia [142,157], California [37], Chile [158], China [54], Croatia [159], France [150,160,161], Mexico [162], Portugal [163], South Africa [164], and Spain [165]. Another important species involved in Botryosphaeria dieback is *Neofusicoccum parvum*, considered one of the most virulent and fastest wood-colonizing fungi on grapevine [45,162,166]. *Neofusicoccum luteum* was also frequently isolated from diseased vines in Australia [167], California [45], Croatia [159], Spain [168], New Zealand [169], Uruguay [170], and Tunisia [171]. Other *Botryosphaeriaceae* species have been found in vineyards in Italy, USA, Germany, Hungary, and Lebanon [11,14,150,163,164].

#### 2.3.2. Symptoms

Botryosphaeria dieback is frequently identified due to the lack of spring growth from affected spurs or due to the shoot and trunk dieback or bud and xylem necrosis [1,149]. In the case of Botryosphaeria dieback, pycnidia develops in dead (cankered) wood [1]. Wood symptoms usually begin in pruning wounds [37,144], and the main wood symptom of Botryosphaeria dieback is a wedge-shaped perennial canker (indistinguishable to that of Eutypa dieback) or circular to non-uniform central staining observed in cross-sections of affected wood [1].

While some studies state that in the field, Botryosphaeria dieback can be distinguished from Eutypa dieback by the lack of foliar symptomatology [1,146,162,172,173], other studies present two forms of the disease: a severe form and a mild form, leading in each case to premature leaf fall. The severe form is characterized by dieback of one or more shoots, accompanied by leaf drop, shriveling, and drying of inflorescences or fruit clusters with a few leaves, frequently, remaining clinging to the shoot tip [150]. Sometimes, new growth occurs from axillary buds in the proximal portion of the shoot. In severe cases, infected shoots die entirely. The mild form is characterized by wine-red (red cultivars) or yellowish-orange (white cultivars) spots on the margins of the leaves or on the leaf blade, which form large zones of deterioration between the veins and the margins of the leaf. Inflorescences or fruit clusters may wither. Leaves that do not drop show various patterns of necrosis [150,160].

Berry rots could be an important Botryosphaeria symptom in various parts of the world and could play an important role in the epidemiology of these pathogens and as inoculum sources for wound infections leading to trunk diseases [151]. Given that pathogens can be found in the wood but not in the leaves of infected plants, it was proposed that the observed leaf and berry symptoms could be actually caused by extracellular compounds produced by fungi in the discolored tissues of the trunk that translocate to the leaves via the transpiration stream [11,27,28].

The progression from infection to symptom onset is chronic and accumulates over time, resulting in production reduction, economic losses, and even vineyard disruption. In the worst case, it leads to the need to replant the entire vineyard long before the normal useful life of the vineyard is reached [7,11,25,37,174,175]. Symptoms can appear in the field only 1 or 2 years after infections occurred [1,172,173], and they are mainly observed in mature vineyards—over 8 years old. However, cankers, dieback, and even plant death have been recorded in 3- to 5-year-old table grapevines [1,162].

#### 2.3.3. Management

Various studies show that grapevine cultivars and rootstocks have different levels of susceptibility to *Botryosphaeriaceae* species [13,48,153,157,166,176,177]. Therefore, the use of tolerant cultivars could be the safest, easiest, least expensive, and most effective way of controlling the disease. Nevertheless, to date, there is no grapevine cultivar registered as tolerant to *Botryosphaeriaceae*. Thus, the general preventive control practices such as the use of healthy viticultural material, limitation of the source of inoculum by removing and burning affected vines, and avoiding strong wounds at spring pruning remain critical to minimize yield losses caused by Botryosphaeria dieback [168].

### 2.4. Phomopsis Dieback

Otherwise known as excoriosis, this cryptogamic GTD affects all viticultural plants in temperate regions, with higher intensity in areas with excessive humidity [178,179]. This disease has been the source of a certain amount of controversy, much of it concerning the identity of the causal agent [180]. Nowadays, some confusion still exists, mainly due to the overlaps of the GTDs whose symptoms are noticed by growers long time after the fungal infection occurs. In the past, excoriosis was wrongly associated by the Anglo-Saxons with black dead arm GTD, and Moreover, the symptoms were frequently attributed to Eutypa dieback or Esca because the fungal pathogen responsible is frequently isolated in the wood of grapevines affected by this GTDs [179,180,181].

Before 1925, this GTD of European origin, first identified in France in 1853 by Fabre and Dunal [182], was known under the name of “punctate anthracnose” because of the strong similarities observed between its symptoms and those caused by anthracnose. [178,181]. The term “excoriose” was introduced in 1925 by Ravaz and Verge [183], for a disease that they observed in several regions of France [180].

In the original description of the disease, Ravaz and Verge [183] considered *Phoma flaccida* to be the cause of excoriosis, but since then the fungus, its name, and various synonyms, as well as its pathogenic ability, have been the subject of considerable debate [180]. For many years, *Macrophoma flaccida* was thought to be responsible for excoriosis in Europe while a similar disease in the USA was attributed to *Phomopsis viticola* [180]. The two diseases were sometimes referred to as European and American excoriosis [180,184,185]. Following reports of *Phomopsis viticola* in Germany, this fungus was considered to be the cause of excoriosis and dead arm throughout Europe, and the pathogenic ability of *Macrophoma flaccida* was put in doubt [180]. Subsequently, the disease caused by *Phomopsis viticola* became known as “Phomopsis cane and leaf spot” [186], and this name has been adopted in most grapevine growing countries. The name “excoriosis” was then rarely used [180].

After many years since the first signaling of this GTD, due to the fast and broadspread in the majority of wine-growing areas around the world, with a more or less sever degree of attack, many scientists were incited to study excoriosis throughout the 20th century. Nowadays, this controverted GTD is studied under the name *Phomopsis dieback,* and it is well accepted that it is caused by *Phomopsis viticola* (teleomorph: *Diaporthe ampelina*) [1,181].

#### 2.4.1. Symptoms

*Phomopsis viticola* is able to attack all the green and growing organs of the vine. Phomopsis dieback is especially known for the symptoms it causes on young shoots in the weeks following bud break and on the vine shoots. The lesions observed on these organs may be different depending on their stage of development [1,179]. Lesions appear on the first internodes at the base of young shoots, often at the level of the lenticels. They are of limited size, elongated in shape, and initially dark in color. They turn brown then black with blue reflections. These necrotic lesions frequently become cankers—the central tissues of the cortex thinning and cracking. This last phenomenon is amplified during a period of rapid growth of the vine. Similar lesions can be observed on the petioles and peduncles of inflorescences and racemes. These can lead to dieback of the bunch, which is no longer fed, or weakening [181,187,188]. In humid conditions, the numerous spots present on the young shoots converge, thus presenting broad, more or less extensive and suberized areas, alternating brown areas and lighter areas of suberized tissue. Some more or less strangled shoots, weakened by the presence of basal lesions, can eventually break or dry out under the action of the wind or the weight of the bunches of grapes. As the tissues age, the healing lesions become increasingly corky, and once the shoots are hardened, the bleaching of the wood can be observed in the fall in addition to necrotic lesions. Moreover, many buds can be infected by the mycelium of the fungus, and they will not emerge the following spring [180,181,189].

Tiny irregular to circular, chlorotic, dark brown to black lesions, often showing a more or less marked yellow halo, also develop on the leaves. The stained leaves become deformed and eventually sifted. In the presence of numerous leaf lesions, large areas of the blade turn yellow, wither, and leaf drop can be observed. Grape berries are also affected. They gradually turn brown and shrivel [181,190,191]. Infections on fruit directly decrease yield and fruit quality [192]. Whatever organs are affected, black punctures pycnidia can be observed in damaged tissues and in the epidermis of bleached wood.

The incidence and severity of *Phomopsis dieback* can vary greatly from season to season [193,194]. Seasonal differences in disease generally have been attributed to variability in environmental conditions such as rainfall and temperature [1,179,180,187,195].

#### 2.4.2. Fungi Involved

*Phomopsis viticola* overwinters in lesions or spots in the buds, bark, and canes of vines infected during previous seasons. Spores are produced in black spots (pycnidia) on the bleached cane. In the spring, the cool, wet weather causes spore release and infection. At least 10 h of rain at 16–20 °C is favorable for spore production. For infection to occur, 6–8 h of leaf wetness are sufficient. Prolonged leaf wetness increases the severity of the disease. Spores on the cane are spread by splashing rain droplets to developing shoots, leaves, and clusters. Usually, the leaf spots (brown with a yellow halo) appear about 21 days after infection, and stem symptoms can take 28 days or more. Shoot infection occurs, most likely, during the period from bud break until shoots are six to eight inches long [196,197].

The fungus does not appear to be active during the warm summer months (above 30 °C), but it can become active during cool, wet weather later in the growing season. Pycnidia eventually develop in infected wood and will provide the initial inoculum for infections during the next growing season. Infected canes and rachises do not produce additional inoculum during the same growing season in which they were infected [196,197].

#### 2.4.3. Management

Selective pruning and protective fungicide applications are commonly used as an attempt to control the disease [192,198]. Mancozeb or captan have been reported as the main fungicides for disease management, applied on a 7- to 10-day calendar-based schedule, [192,198,199,200]. Many growers begin the fungicide application schedule when the first few leaves start to expand, and shoots are 10–20 cm long. Growers wait until that growth stage because it seems uneconomical to apply fungicide onto smaller tissues [192].

### 2.5. Eutypa Dieback

Eutypa dieback, also referred to as eutypiosis [11], is a major wood canker disease identified in perennial agricultural crops, including grapevines (*Vitis* spp.) worldwide, most frequently found in vineyards that register more than 250 mm of rainfall per year [201]. The chronic infections of the wood lead to significant economic losses due to cumulative yield losses, increased crop management costs, and shortened life span of the vines [9,202,203,204]. The disease is progressive over many years, and if it progresses unchecked, infected vines may die within 10 years of inoculation [205,206,207,208].

#### 2.5.1. Fungi Involved

The attribution of the name to the agent responsible for branch dieback is ambiguous. Pathogenicity of *Eutypa* sp. first was reported on apricot, and the causal agent was named *E. armeniacae*. However, no morphological differences were reported with the previously described *E. lata*, and some authors considered both species synonymous. Others regarded them as distinct species on the basis of pathogenesis and molecular analysis [209].

Although *Eutypa lata* is well accepted as the primary causal agent of Eutypa dieback, other diatrypaceous species such as *E. leptoplaca*, *Cryptosphaeria pullmanesis*, *Cryptovalsa ampelina*, *C. rabenhortsii*, *Diatrype* sp., *Diatrype oregonensis*, *D. stigma*, *D. whitmanensis*, *D. vulgaris*, *Diatrypella verrucaeformis*, *Eutypella vitis*, *E. leprosa, E. citri-cola*, *E. microtheca, and E. scoparia*, are reported from grapevines with Eutypa dieback symptoms worldwide [153,204,210,211]. Eutypa lata can however be recovered from the wood lesions alone or in combination with other fungi such as *Phaeomoniella chlamydospora*, *Phaeoacremonium aleophilum*, *Sphaeropsis malorum*, *Phomopsis viticola*, and *Phellinus ignarius* and with its antagonist *Gliocladium roseum* [212,213,214].

*Eutypa lata* has been identified as the causal agent of Eutypa dieback in major grape-production regions, including California, Europe, South Africa, and Australasia [204,209,215,216]. Within these regions, *Eutypa lata* is reported as the predominant cause of canker and dieback of grapevines in the northern and coastal regions (annual precipitation of at least 3.5 cm) but is less common than other GTD pathogens in the dryer and hotter southern and central areas of the state [173,204,217]. Symptoms of Eutypa dieback have also been described from grape-growing regions of eastern North America, specifically the states of New York, Michigan, and Ontario (Canada) [204]. Recent investigation suggests the fungus to be of European origin, and the current global distribution is likely a result of the multiple introductions of genetically diverse genotypes into new areas (North America, Australia, and South Africa) via transport of infected plant material [218].

This fungus produces perithecial stroma on diseased grapevine wood [201]. Ascospores are released and disseminated throughout the entire year, with each rainfall greater than 0.5 mm. Their liberation begins 2–3 h after the onset of rain and stops 24 h after the rain stops. Ascospores penetrate the plant by infecting susceptible pruning wounds during winter dormancy [11,219,220,221].

#### 2.5.2. Symptoms

Several years following the initial wound infection, a wood canker develops, and dieback becomes apparent. The external characteristic symptoms of Eutypa dieback are most conspicuous after bud break and include stunted shoots with shortened internodes, formation of small, deformed (cupped and tattered) chlorotic leaves, and development of small and straggly fruit clusters [204,218,222]. Most of the flowers dry before opening. Wood symptoms are given by internal necrosis, seen as a wedge-shaped area of stained tissue if a cross section is made of an infected trunk or cordon, and external cankers form around sites of infection [208,215,223]. The mycelium decays the wood, in part, through production of cell-wall–degrading enzymes [218,224]. The characteristic misshapen, dwarfed leaves are attributed to the translocation of phytotoxic fungal metabolites, via the vascular system, from infected wood to the herbaceous part of the plant [218,225,226].

#### 2.5.3. Management

Management of *Eutypa dieback* relies mainly on the sanitation of the vineyard by removing the dead wood, avoiding pruning during rainfall events, and/or delaying pruning until later in the season, and avoiding the treatment of the wounds with fungicides or biological control agents [208].

There are several field trials that showed Benlate’s efficacy in preventing Eutypa dieback [208,227,228,229,230] thus Benlate (DuPont de Nemours & Co., Wilmington, DE, USA) was registered for *E. lata* control for 30 years [230]. Biocontrol agents *Bacillus subtilis* [231], *Fusarium lateritium* Nees:Fr., and *Cladosporium herbarum* (Pers.:Fr.) Link [232] were also tested as an alternative method for control of *E. lata* and showed some potential activity in limiting the establishment of the pathogen [230].

Boron was also found to be active against several wood decay fungi [230]. Irelan et al. [233] showed the efficacy of boric acid treatment to control infection of pruning wounds by *Eutypa lata* in field trials. Moreover, Rolshausen and Gubler [230] demonstrated the fungicidal activity of boron against *E. lata*. In their study, boron-based products resulted in over 75% disease control 10 to 12 days following treatments [230].

### 2.6. Esca Disease

Esca disease is one of the most complex and important vascular diseases belonging to the GTDs cluster with worldwide spreading [234,235]. It is particularly harmful due to the fact that it is very widespread, it causes direct losses in production (lowering the quantity and quality of the grapes), it significantly reduces the lifespan of the vineyard, and also because of the lacking measures to control or limit the disease [235,236,237,238].

Esca is the GTD with the oldest history. Although it was described since the ancient Greek and Latin cultures [178,234,239] and then in the Middle Ages [27], the related research in etiology began only at the end of the 19th century with Ravaz [240] and Viala’s [241] work [234].

#### 2.6.1. Fungi Involved

A broad range of taxonomically unrelated fungal trunk pathogens [40,58,238,242,243,244,245] and even endophytic bacteria [246] have been isolated from wood tissue of Esca deceased vines [1]. *Phaeomoniella chlamydospora*, *Phaeoacremonium aleophilum,* and *Fomitiporia mediterranea* are considered the main causal agents [11]. *Phaeomoniella chlamydospora* and *Phaeoacremonium aleophilum* were identified on grapevines in France [247], Spain [248], Montenegro [249], Iran [250], and California [251]. *Fomitiporia mediterranea* was reported in Italy [252]. *Togninia minima* was also reported as an esca pathogen in California [43]. *Stereum hirsutum* could also play roles in the esca disease complex [50,100,253,254]. The role of these microorganisms and their way of interaction is still uncertain. The main hypothesis is that the young vines with Petri Disease, infected with the pioneer fungi *Phaeomoniella chlamydospora* and *Phaeoacremonium* species, can later develop Esca symptoms following further colonization by several basidiomycetous species belonging to the genera *Inocutis*, *Inonotus*, *Fomitiporella*, *Fomitiporia*, *Phellinus,* and *Stereum* [1,245,255]. The mechanisms of Esca pathogenesis are still largely misunderstood [11,245].

#### 2.6.2. Symptoms

This latent disease primarily affects perennial organs (the trunk), causing necrosis of internal tissues. Annual organs (leaves and clusters) typically begin to display symptoms in mature plants—older than 10 years [49,256,257].

Generally characterized by the development of typical inner necrosis in grapevine wood tissues and external symptoms known as “tiger-striped” leaves or black measles on the berries, Esca (like the rest of the GTDs) is assigned to infection by pathogenic fungi that invade the perennial plants and their vascular systems [27,50,245,258].

Based on the views of many authors, Esca is described as having two forms: a chronic/mild form (also referred to as grapevine leaf stripe disease—GLSD), characterized mainly by discolorations and/or scorching on the margins and/or in between the veins of leaves (producing an overall tiger-striped appearance on the affected vines) and the acute/apoplectic form that is characterized by a sudden and severe collapse of the entire vine [1,11,27,49]. Other authors describe Esca either as a complex disease, in the sense that a number of interacting factors and various microorganisms acting together are determining the whole syndrome of Esca [27], or a complex of five distinct diseases: dark wood streaking (seen especially in the rootstock of grafted cuttings); Petri disease, formerly known as “black goo”, “slow dieback”, “*Phaeoacremonium* grapevine decline” (which affects mainly young vines of 2–7 years), young Esca; white rot and Esca proper [53].

Cross sections of the Esca diseased vines trunk generally reveal a variety of lesions and/or decay types [1,237,259]. The wood of older vines usually shows a white to yellow soft rot, from which basidiomycetes are usually isolated [50,260,261,262]. The wood lesions continue to extend as the vines age, and dead tissue increases in volume and can cause, in severe cases, a sudden wilting of the vine known as apoplexy [1,27,237]. The most frequent and characteristic internal symptom of Esca is the sponginess and the softness of the woody tissues [234].

#### 2.6.3. Management

Up to today, there are no registered chemical or biological solutions to control this disease. Studying and developing traditional solutions for mitigating Esca is difficult because of its complexity. When a grapevine shows symptoms of Esca, the recommendation is the removal and destruction by fire. If the plant is considered still viable, options include chirurgical removal of the damaged organs and the retraining of new canes to replace the ablated organ [13,32]. Protection against Esca relies mainly on prophylactic measures and preventive treatments such as pruning wound disinfection [1,21,263]. Di Marco et al. [236] very well pointed out that in order to control Esca GTD, it is necessary to consider the age of the plant, the stage of the infection, and the incidence of the disease in the vineyard, since if the incidence becomes too high the possibilities of control are correspondingly reduced.

Promising results in the control of the disease were obtained with aluminum fosetyl and its main metabolite, phosphorous acid, tested under controlled conditions against the pathogens involved in Esca [264,265]. In addition to reducing *D. seriata* and Esca complex vascular pathogens in wounds, fosetyl-Al (organic salt) was effective at reducing the symptoms of the Esca complex foliar disease (Esca proper) [266,267]. Moreover, on the Esca complex vascular pathogens, the effectiveness of the phthalimide captan [62,268] and dodecyl dimethyl ammonium chloride [40,62,268] was confirmed.

In terms of testing natural active ingredients, chitosan inhibited the mycelial growth of Esca complex fungi [21]. Brown seaweed extract with CaCl2+Mg (NO3)2 applied topically to vines with Esca decreased the frequency and intensity of foliar symptoms [269,270,271]. The vascular pathogens implicated in the Esca complex were examined in vitro when treated with the phytoalexins resveratrol, p-coumaric acid, and pterostilbene. The findings varied depending on the phytoalexin and the pathogen. Pterostilbene inhibited every pathogen that was tested [272], but resveratrol displayed variable outcomes depending on the pathogen [272,273].

The effectiveness of a mixture of seaweed extract and inorganic salt in reducing the symptoms of Esca, as well as a mixture of garlic, chitosan, and vanilla for wound protection, as natural bioactive compounds, points to the possibility of developing natural-based tools to aid and limit losses caused by GTDs in organic viticulture [269,274]. The results of biostimulants up to this point, however, indicated that they are mostly ineffective in vineyards and occasionally cause an increase in disease incidence, probably as a result of faster and/or larger transport of fungal poisons to the leaves [21].

Moreover, several control trials were conducted on individual diseased vines to assess the effectiveness of treatments with Triazolic fungicide and wound protection with *Trichoderma viride* applications [236]. Pruning wound treatment using Trichoderma can be both preventative and long-lasting. To prevent the spread of GTD infections in new vineyards, it is crucial to treat pruning wounds as soon as possible with Trichoderma-based remedies. The best Trichoderma strains should be chosen in accordance with the grapevine variety and the environmental conditions. Being the sole method available at the moment for reducing the in-field Esca complex, Trichoderma species and strains included in registered products actually cause a noticeable reduction in the foliar symptoms [275]. It is worth noting that starting in the second or third year of a multi-year treatment of a vineyard, a *T. atroviride* strain I-1237 and a *T. asperellum* and *T. gamsii* mix (Remedier) reduced the incidence of and mortality in Esca complex impacted vineyards [275].

*B. subtilis* decreased the prevalence of the vascular pathogens linked to the Esca complex (Petri disease) in nurseries, but the severity of internal symptoms rose [68]. The rhizospheric *Pythium oligandrums* enhanced plant defenses against Esca pathogens and demonstrated significant persistence in the root system [21]. *P. oligandrum*, according to Yacoub et al. [276], induces a physiological state known as priming, which enables the plant to deploy its defense mechanisms more vigorously in response to infection by the Esca complex pathogens [276].

All of this research unequivocally shows that there are numerous current potential solutions, even if none can be considered an easy tool to add to the GTD management procedures just yet [21].

## 3. Viticulture in Romania

Romania, located between 43°370′–48°150′ N latitude and 20°150′–29°440′ E longitude, is part of Eastern Europe and has a continental temperate climate with four seasons. The climate is also influenced by the steppe climate in the east, the Mediterranean climate in the south-west, and the oceanic climate in the west and north-west [3,277]. In 2020, the annual average temperature was 10.8 °C with maximum annual average temperatures of 22 °C to 24 °C (in the summer) and minimum annual average temperatures between −3 °C and −5 °C (in the winter) [278].

The topography of Romania is diverse and distinct. It comprises 28% mountains, located in the middle of the country, 42% plateaus and hills, and 30% plain land [3,277]. More than half of Romania (62%) is covered by arable land [277,279]. The soils are fertile containing mostly chernozem (humus rich black soil), and on higher elevations, they change to podzolic soil [280]. The grapevine harvested areas are mainly concentrated in hilly and plateau areas [3,277].

Due to the adequate climate for grape production and the fertile soils, in Romania, viticulture is a traditional practice that has arisen and developed throughout history [4,277]. Nowadays in Romania, there are eight large grapevine growing regions (Table 1) with specific environmental conditions, as a result of their proximity to the Carpathian Mountains (2500 m altitude), the Danube River, and the Black Sea [277]. These regions include 141 vine plantations [277] from 37 main vineyards—the largest number of vineyards in the European Union—82% of them being cultivated with grape varieties used for wine production [4]. The wine production types are different from south to north depending on the accumulation of heliothermic resources, due to the wide latitude of the Romanian land [277].

## 4. Grapevine Trunk Diseases in Romania

Diseases and pests of grapevine became a major concern throughout Europe following the introduction of grapevine parasites from the American continent in the second half of the nineteenth century. Until the 19th century, in Romania like in other European countries, parasitic diseases of the vine were considered mostly a divine punishment [178]. After that, following a sharp decline of vines in many vineyards due to high mortality of the vines, the attention of researchers in Romania has been focused on preventing and combating what have become known as cryptogramic diseases [178]. On this matter, researchers like Crişan [282], Mărmureanu [283], Rafailă [284], Oprea [285], Podosu [286,287], Tică [288], and Ulea [178,289] have revealed some species of pathogenic fungi specialized in the degradation of vines bark and wood, the most important being: *Eutypa lata* [178,285,290,291,292,293,294,295], *Phomopsis viticola* [178,285,288,290,291,293,295,296,297,298], *Stereum hirsutum* [178,285,290,291,293,295], and *Phellinus igniarius* [178,285,290], nowadays known, in Romania, as the causal agents for Eutypa dieback, Phomopsis dieback (Excoriosis) and Esca. To a lesser extent *Phaeoacremonium aleophilum* [297], *Phaeomoniella chlamydospora* [297], *Nectria destructans* [178,290], *Cylindrocarpon destructans* [290,292], *Diplodia seriata* [295], and other *Botryosphaeriaceae* [292] were also identified in Romanian vineyards, and these are, nowadays, known as causal agents for Petri disease [295]. Blackfoot disease [178], and Botryosphaeria dieback.

At the present time, in Romania, there is quite limited scientific information available regarding the fungal GTDs. Studies on this matter are scattered and unpublished in international digital databases, being available only in Romanian language on paper. Researchers interest in this subject was given by the fact that during the decade 1983–2006 the grapevine decline turned into calamity, leading to more than 50% early drying of the country’s grapevines. Since then, this phenomenon has subsequently diminished, due to tolerant cultivars and superior culture technologies (nurseries certification, planting practices assuring that the planting young vines are certified as healthy and free of GTD, and pruning the old vines by cutting and burning all the diseased and dead wood), and in 2008, the phenomenon of grapevines early drying was reported as restrained to 5–35% [299].

Due to the increasing importance of GTD, different studies were conducted in order to understand the factors influencing the grapevine infection with the GTD fungal pathogens. Comșa et al. [293] conducted GTD observations on grapevine cultivars representative for the Târnave vineyard (VZ1): Fetească regală, Muscat Ottonel, Italian Riesling, and Sauvignon blanc. They monitored the frequency and intensity of the infections caused by the main lignicole pathogenic fungi *Eutypa lata*, *Phomopsis viticola,* and *Stereum hirsutum* [293]. The first signs of the decline syndrome were noticed in spring when the grapevines entered the leaves development phenophase [293].

The most recent study of GTD pathogens identification in some Romanian vineyards evaluated bark, canes, and trunk wood collected from Blaj-Târnave (VZ1), Ciumbrud-Aiud (VZ1), Cuvin-Miniș (VZ5) and Sarica Niculitel (VZ6) vineyards [295]. In this work, the kind and color of mycelial hyphae, as well as the shape, color, and size of spores, were used to identify fungal species using macroscopic and microscopic examinations of isolates from Petri dish culture on Sabouraud medium [295].

As the literature review proves, the most important GTD in Romania are the Eutypa dieback, Phomopsis dieback (known as Excoriosis), and Esca. Based on the frequency of pathogenic fungi associated to vine dieback in Romanian vineyards, Eutypa dieback ranks as the most frequent GTD, *Eutypa lata* being identified in 32% [290] up to 52% of the analyzed samples [292]. Phomopsis dieback ranks as the second most frequent GTD in Romanian vineyards, with *Phomopsis viticola* being identified in 21% [292] and up to 27% [290]. In Romanian vineyards, Esca is less frequent, being identified in 4% [178,292] and up to 8% [178,291] of the analyzed samples. The main symptoms of the three GTDs observed in the field in Romanian vineyards are presented in Figure 3, and they will be detailed along with each of the GTD. 

### 4.1. Eutypa Dieback in Romania

In Romania, *Eutypa lata* was first mentioned by Rafailă and Oprea as found in the vineyards of Comarna, Cotnari, and Bucium, in Iași County (VZ2), in 1982 and later in other vineyards all around the country: Miniș (VZ5), Diosig (VZ5), Jidvei (VZ1), Dealul Mare (VZ3), Valea Călugărească (VZ3), Odobești (VZ2), Murfatlar (VZ6), Ostrov (VZ7) [178,285], Pietroasele (VZ8), Alba (VZ1), and Apold (VZ1) [284,285,294,295,300] (Figure 4).

#### 4.1.1. Risk Factors

*Eutypa lata* infection associated with ecological weakening factors determined the grapevine’s dieback all over Romania’s vineyards [291]. The percentage of infested grapevine trunks and canes was between 3% and 96%; the disease was found in vineyards older than 6 years of age, especially on the ones of 10–15 years old [291]. Thus, the age of the plantation has a significant influence on the presence of Eutypa dieback [294]. This does not necessarily imply that young plantations are less vulnerable to *Eutypa lata* infection; the absence of symptoms is attributable to the pathogen’s delayed evolution in grapevine wood [294]. The disease is found with a much higher frequency in neglected vineyards, where the phytosanitary measures were not applied according to the recommendations of the specialists [301].

No resistant cultivars to the attack of *Eutypa lata* fungus are known [302,303]. Oprea and Podosu [291] present in their study the most sensitive grapevine cultivars as being the ones for wine, especially Italian riesling (in Diosig—Bihor and Gaiceanca—Bacău vineyards) where on certain plots, all vine trunks and canes showed specific symptoms of *Eutypa lata* infection [291]. Other cultivars very much affected by the disease were: 60% Cabernet Sauvignon, over 30% Fetească albă, between 10 and 89% Băbească neagră (much severe in Nicoreşti—Galaţi vineyard), about 63% Fetească regală, 30% Muscat Ottonel (especially in Miniș vineyard), and between 3 and 33% of the autochthonous cultivars: Grasă de Cotnari, Galbenă de Odobești, and Șarbă [291]. The most infected grapevine cultivars for table grapes were Chasselas d’Oré (68% in Valea Călugărească vineyard) and Afuz Ali (in south Romania vineyards) [285,291,303].

In Târnave vineyards Fetească regală, Fetească albă, Sauvignon blanc, and Muscat Ottonel cultivars were assessed for the Eutypa dieback infection evolution in the vine plantations of different ages: young vineyards of 6 years and old vineyards over 25 years [294]. The attack degree (AD) was found to be substantially lower (1.5%) in young plantations compared to older ones (18%), regardless of the cultivar [294]. In terms of studied cultivar, independently of the plantation age, Feteasca regală (7.0%) and Sauvignon blanc (7.3%) had the lowest AD, followed by Feteasca albă (8.7%) and Muscat Ottonel (15%) as the most sensitive to *Eutypa lata* in the microclimate of Târnave vineyard.

#### 4.1.2. Symptoms

Overall, the *Eutypa lata* fungus produces metabolic disorders in the vine, and the characteristic symptoms of Eutypa dieback appear under the influence of the released toxins. The winter climatic conditions (low temperatures and precipitation quantities) may influence the characteristic symptoms expression. When the disease has a powerful expression of symptoms, during the start of the vegetation period of the grapevine, a proliferation of shoots is observed, and they continue to have an abnormal development for the rest of the vine’s life [285]. The leaves remain tiny, chlorotic, and distorted with ragged borders [178,285,293,303]. The inflorescences may wither after flowering or form clusters of beaded berries [178,285]. The first-year canes do not develop normally; they remain short, thin, and with an obvious short knotting. Oprea and Dumitru [285] state in their work that the normal autumn canes measure 93.8 cm on average, compared to the annual canes of the vines affected by Eutypa dieback that measured between 19.55 and 22.3 cm. In autumn, the fruit clusters on the affected vines remain very small. Oprea and Dumitru [285] show an average weight of the fruit cluster of 5.8–12.8 g on an affected vine compared to 72 g of a healthy, normal fruit cluster. In sections, the affected arm presents specific wedge-shaped necrosis, in the xylem area, where the primary infections took place. Phloem and cortical tissue, secondarily invaded, form ulcerative lesions. In a more advanced stage of the disease, the fungus invades and destroys the secondary walls of the libero-ligneous vessel; the tissue becomes very frail, and thus, the wood can easily break on bending [178,285,291]. Cross sections of the affected wood present distinct, hard V-shaped necrosis, with the margins of contaminated areas turning brown-gray to brown-red/purple depending on variety [291,293].

#### 4.1.3. Fungi Involved

Eutypa dieback is propagated, from year to year, mostly by *Eutypa lata* ascospores that grow in the perithecia of dry wood. They can stay viable for up to five years, which is why Eutypa-infected stocks can be a long-term source of infection [178,285,293]. Infections develop in late autumn and early spring, when rainfall is favorable [293]. Although *Eutypa lata* is the well-known and studied fungal pathogen inducing Eutypa dieback, in the Blaj-Târnave vineyard, on Victoria grapevines, other *Diatrypaceae* sp. were also identified on trunk samples infected with Eutypa dieback GTD [295].

Biological parameters such as temperature, pH values, and humidity, as well as the energy sources, such as carbon and light, may have an influence on the development of pathogenic fungi. On this matter, Oprea and Podosu [291] conducted a study in order to determine the influence of the biological parameters on the development of *Eutypa lata* colonies.

##### Temperature Influence on the In Vitro Development of *Eutypa lata*

The development of *Eutypa lata* colonies on Chloramphenicol Glucose Agar (CGA) medium was strongly influenced by the temperature levels. The fungi colonies start to grow at +8 °C, and they show a white, lax mycelium which is observed as yellow on reverse [291,295]. After 15 days, the colonies reach 20 mm diameter. In these conditions, fructifications were not formed. As the temperature rises, the colonies diameter increases, the mycelium becomes dense and takes a felt aspect. The optimal temperature for *E. lata*’s development was registered between 18 and 26 °C. Black crusts developed on the mycelium surface, and gray pycnidia appeared due to the abundance of hyphae. The maturation of pycnidia takes 12–14 days and is contained in a gelid mass that is eliminated via ostiols. Perithecia continued to be formed in several of the cultures incubated at 18 °C [291].

The highest temperature for *Eutypa lata*’s development was found to be 34 °C, and 36 °C may be considered the lethal level for this fungus. When the temperature dropped from 36 °C to 22 °C, the colonies on Petri plates continued to produce mycelium. The temperature also has an important role in determining the germination of *Eutypa lata* spores. Pycniospores do not germinate on the medium culture (water, agar, or maltose 5%) in the amorphous stage unless they are not exposed to UV rays for 15 min. Germination began two hours after irradiation at a low percentage (2%), and the length of the germination tube was 1.5–2 µm. The pycniospores germination percentage grew with the duration of the irradiation exposure time, and as a consequence, after 24 h, spore germination was 46%, and germination tube length was 52–65 µm. In the typical conditions of water–agar medium, ascospores present in the teleomorph stage germinated. Germination began at 10 °C and continued until 30°C; the ideal temperature was between 18 and 28 °C [291].

##### Relative Atmospheric Humidity (RH) Influence on the In Vitro Development of *Eutypa lata*

Besides temperature, the development of *Eutypa lata* fungal colonies is controlled by relative air humidity levels, according to observations made in vitro. Mycelium development was halted by humidity levels below 30.5%, but it became abundant without fructifications at RH values between 39% and 74%. When the humidity level exceeds 80%, morphological changes occur. The fungus fructified and produced pycnidia after 4 weeks that were matured after 7–11 days [291].

##### Influence of pH Values on the In Vitro Development of *Eutypa lata*

*Eutypa lata*, the observed lignicoulus fungus, grew in an acidic (4) to basic (11) pH growth media. The fungus displayed limited development colonies with a low vegetative mass at the acid level (4). Fungi colonies with a rich vegetative mass and pycnidial fructifications were formed from low acid to strong basic pH (5.5–11) [291].

##### The Energetic Resources of *Eutypa lata* Growing In Vitro

(a)Carbon source influence

The findings of the in vitro experiments revealed that carbon was the most critical factor for the formation of the lignicoulus fungus colonies. This element, a component of the carbohydrate molecules, can be assimilated in various ways depending on the source and the carbon chemical link in that molecule. Carbon is hydrolyzed from monosaccharides such as glucose, fructose, trehalose, ribose, arabinose, levulose, and rhamnose. Melibiose and sucrose are the hydrolyzed disaccharides, and cellulose and lignin are the polysaccharides ones. The colonies developed on a growth medium which included these carbon resources, grew to their fullest size, and fructified well. On sorbosis growth medium, as well as on galactosis, maltosis, and inulinase media, *Eutypa lata* developed more slowly. The fungus thrives on manitosis, galactosis, levulosis, melibiosis, and starch culture medium; however, it does not fructify. Colonies of *Eutypa lata* fungi were unable to grow in the absence of carbon [291].

(b)Light influence

The CGA growth medium on Petri plates containing fungal colonies was exposed to a constant source of light, light/dark alternating (8 h with 16 h or 12 h with 12 h), and continuous dark. The fungus colonies that were kept in a constant dark environment had the best vegetative mass; the colonies had a dense, felt-like appearance. However, weak fructifications were developed. In a light dark alternation of 8 to 16 h, *Eutypa lata* fructified very effectively [291].

### 4.2. Phomopsis Dieback in Romania

In Romania, the first report of Phomopsis dieback (well known as excoriosis) was made by Săvulescu et al., in “Starea fitosanitară” [Phytosanitary Status -RO] in 1926–1929 and 1932–1933, but they attributed the disease to other pathogen agents (*Phoma flaccida*, *Phoma reniformis*). This confusion has lasted for some time, as it did in the rest of the world, many authors reporting excoriosis symptoms attributed to other fungal species. [178,303]. *Phomopsis viticola*, the etiological agent of grapevine excoriosis, was first identified in Romania in 1962 by Crișan. In 1970, Rafailă reported the GTD in various Romanian vineyards describing characteristic symptoms and connecting for the first time the excoriosis with *Phomopsis viticola* Sacc. Due to the expansion of the affected area and the damage caused, especially after 1967–1968, many Romanian researchers, such as Oprea and Dumitru [285], Mărmureanu [283], Tică [288,298], and Ulea [178,289], have studied excoriosis. All authors highlight the practical importance of this GTD, which causes significant losses in some of the Romanian vineyards, such as Iași (VZ2), Cotnari (VZ2), Dealul-Bujorului (VZ2), Odobești (VZ2), Murfatlar (VZ6), Jidvei (VZ1), Alba (VZ1), Ciumbrud-Aiud (VZ1), Miniș (VZ5), Arad (VZ5), Dragășani (VZ3), Bihor (VZ5), Valea Călugărească (VZ3), Pietroasele (VZ3), Panciu (VZ2), and Ostrov (VZ7) [178,294,295,296,298,300,304] (Figure 5).

Ulea [178] reports excoriosis in Moldova’s vineyards during 1968–1992 as follows: in 1982 in Cotnari and Iași vineyards (VZ2) at a 0.01% attack degree (AD), in 1983 in Iași and Odobești vineyards at 0.25% AD, again in 1985, in Odobești vineyard at 0.25% AD, in 1990 in Iași vineyard at 1.5% AD, also in 1991 in Iași and Odobești vineyards at 3% AD and in 1992 in Iași vineyard at 0.01% AD [178]. In the Odobești vineyard, the following cultivars were detected with this GTD: Fetească regală, Italian Riesling, Șarbă, Furmint, Chasselas d’Oré, Galbenă de Odobești, and Plăvaie. Feteasca regală, Italian Riesling, Șarba, and Chasselas d’Oré cultivars had excoriosis also in the Panciu vineyard, and besides these, the cultivars Rkatiteli and Aligote. In the Cotești vineyard, *Phomopsis viticola* infected Fetească regală, Italian Riesling, Șarbă, Chasselas d’Oré, Aligote, Plăvaie, Fetească albă, Hamburg Muscat, and Afuz-Ali cultivars [304]. In the Târnave vineyard (VZ1) the *Phomopsis viticola*’s AD was evaluated in May, and this parameter was 28.06% for Fetească regală, 27.86% for Italian Riesling, 22.50% for Sauvignon blanc, and much lower (of 5.70%) for Traminer roz [297]. *Phomopsis viticola* was found to cause excoriosis on the Traminer cultivar also in the Aiud-Ciumbrud vineyard (VZ1) [295].

According to Oprea and Podosu [291], grapevines growing in places with a clay compact acid soil, excessive watering, and industrial pollution are more impacted by excoriosis [291]. In Vrancea vineyards (VZ2), the loss rate in vineyards located in river meadows, on low, cold, and humid terrains, reached 35% [304]. It was also reported that the behavior of grapevine cultivars in response to this disease under natural infection conditions varies depending on the pedological and climatic characteristics of the vineyard. However, due to their genetics, some cultivars like Fetească regală, Sauvignon blanc, and Italian Riesling are more susceptible to this fungal disease than others [297].

Tică et al. [288,298], by comparing the behavior of the grapevine cultivars to the attack of *Phomopsis viticola* in the conditions of Vrancea vineyards (VZ2) (Odobești, Cotești, and Panciu), conclude that in conditions of natural infection, the susceptibility of the cultivars may differ for the same cultivar depending on the pedoclimatic and microclimatic conditions of the vineyard. Thus, the Plăvaie cultivar oscillated between very sensitive and sensitive (AD = 16.6–52.4%), the cultivars Galbenă de Odobești and Furmint between medium-resistant and sensitive (AD = 6.6–17.3%), and the cultivars Fetească regală, Fetească albă, Italian Riesling, and Șarbă between resistant and sensitive (AD = 1.3–14.2%). The Afuz-Ali cultivar behaved as sensitive (AD = 12.3–22.2%). The cultivars Merlot, Frâncușă, Chasselas d’Oré, Aligote (AD = 0.1–1.3%), and Muscat de Hamburg (AD = 1.5–2.2%) proved to be tolerant [298,304]. From the data obtained in the experiment but also from the data in the specialized literature, the Merlot cultivar showed tolerance in Vrancea, in the Dealul Mare vineyard, and in those in France, while the Aligote and Chasselas d’Oré cultivars, which became tolerant in Romania, are sensitive to the pathogen in France [304].

Oprea and Dumitru [285] followed the behavior of grapevine cultivars when attacked by *Phomopsis viticola* in Valea Călugărească (VZ3), and they found that Cardinal, Cabernet Sauvignon, Fetească regală, Muscat de Hamburg, Muscat Ottonel, Mustoasă, and Clairete are very sensitive (AD = 40.31–54.68%). The plots in which the phenomenon of vines drying was more severe were located on flat ground, with soil that maintained humidity [285].

#### 4.2.1. Symptoms

In Romania, the symptoms of excoriosis were discussed in the complex of premature death of grapevines by Oprea and Dumitru [285] and Mărmureanu et al. [283], and they consider that excoriosis is a focal point disease. Its spread is primarily due to human intervention, grafting being the main cause, as a result of the use of infected canes. The damages thus appear from the nursery [285]. In Romania, in the period 1986–1990, Tică et al. [288] investigated the influence of the pathogen *Phomopsis viticola,* present in grafting eyes, on the phytosanitary condition of the vine during the grafting–forcing period and on vines planted in the nursery, in Odobești vineyards. From the data analysis, it was established that the yield of STAS vines decreased in the excoriosis attacked vines group from 31.5% to 15.7% [178,288].

Another aspect studied by the same authors was the influence of excoriosis on budburst. Several cultivars were studied, and the conclusion is that the buds at the base of the shoots are more affected than those located at the top of the shoots—13–81% buds from the 1st to 4th eye that did not start vegetation in spring compared to 6–18% of the buds located at the upper part (7–10th eye). The best start in vegetation was recorded in the last eyes on the shoots (96–98%) [178,288].

Same authors also studied the influence of excoriosis on grape production of various cultivars in terms of quantity and quality. The results show that grape production decreases in the case of infected vines with up to 48.4% (Plăvaie cultivar). The lowest drop of grape production was 5.9% registered on the Chasselas d’Oré cultivar. Galbenă de Odobești, Furmint, Italian Riesling, and Șarbă registered decreases of grape production ranging between 39.7% and 27.2%. These differences were linked to the attack degree and thus to the cultivars’ disease tolerance [178,288]. Podosu et al. [304] also established that during the offshoot growth phase, damage is significant, and the influence of this damage on the quantity and quality of the yield has been assessed.

The sugar content of grapes from infected vines was lower compared to the healthy ones (an average of 165.5 g/L compared to an average of 174.4 g/L). The acidity of the must (grape juice) was not influenced by excoriosis, pH values of the grape juices not showing significant differences [178,298].

The first symptoms of excoriosis appear at the beginning of the vegetation period, when bud break is delayed, under the form of dark spots isolated or associated of about 0.5–2 cm × 0.3–1.0 cm, usually located at the base of canes [288,293,304]. In the Valea Călugărească vineyard, the budburst period was 12–14 days delayed, and the basal buds were dead [304]. As the infection evolves, ulcerations occur, budding is delayed, and the buds at the base of the canes die, leaving only those at the top viable, which pushes the fruit elements higher on the shoots. Black, small, round, or linear, more or less deep lesions appear in the shoots. In Târnave vineyard when the observations were done later (May–July), cane and leaf spots were observed [297]. On the leaves, petioles, and then on racemes and pedicels, during summer, appear circular spots, brown-blackish in the center with a yellow-orange halo. During strong attacks, the shoots remain small, stunted, sometimes in a fan form, and they detach easily. After entering the veraison, the grape berries rot and are, sometimes, covered with the characteristic pycnidia [178,293,298]. In the fall, on the canes, Ulea [178] signaled a pronounced whitening of the bark, especially on the first basal internodes, on which small black pycnidia are defined. At the end, the vine dies in the summer, with the leaves and fruit clusters on it [178,293,298]. The infections are favored by cold and wet weather [293]. For instance, in the Târnave vineyard for the period 2009–2011, *Phomopsis viticola* had a higher intensity in the vineyard in 2010, the wetter and colder year of the interval. This allowed the authors to establish correlations between disease intensity and weather conditions [293].

#### 4.2.2. Fungi Involved

##### Life Cycle of *Phomopsis viticola*

The *Phomopsis viticola* Sacc. fungus survives the winter under pycnids form on the basal arms and as mycelium in the dormant buds [288,304,305]. During this time, the pycnids formed in autumn mature. The pycniospores (α and β of which α spores are the most important source of infection) start to form near the end of winter (in February) and are present in large quantities in spring in the sprouting phenophase [285,288,304] (Figure 6).

Mature conidia (pycniospores) are dispersed by rain drops, and they germinate on the canes in 4 to 8 h depending on the temperatures (25 °C optimal temperature; −15 °C lower limit). After pycnospore germinations, the infection starts in 5 h in optimal temperature conditions (25 °C) [288]. At a temperature range of 15–18°C, the infection begins after 7 to 10 h of continuous humidity of the vine [288,304,306]. The most sensitive to infections are the young offshoots, from the beginning of the shoot growth until the spring rains [304].

##### Biology of *Phomopsis viticola*

Podosu et al. [304] isolated the pathogen *Phomopsis viticola* Sacc. from samples consisting in grapevine arms, shoots and roots, withered or dry, collected from Vrancea’s vineyards, which showed signs of decline. The samples were kept in the humid chamber at temperatures of 18–22 °C. After fructification, the fungus was isolated on a GCA medium and then purified in test tubes or Petri dishes. The in vitro biological aspects sought to determine the influence of some agarized culture media of the GCA culture medium reactions and the effect of temperature and relative humidity on the growth and spore-production of the fungus.

The *Phomopsis viticola* Sacc. isolation was obtained by keeping the contaminated material in a humid container, where the necrosis affected areas began to show the typical pycnids (with white-yellowish to orange cilia) and pycniospores (α and β type). The expansion of the colonies has been observed daily (by measuring their diameters) as well as the spore-production (macroscopically and microscopically) [304]. The results show that *Phomopsis viticola* Sacc. developed in acid (4) to basic (11) pH of the growing medium. At the acid pH level (4), the fungus showed limited growth colonies with a low vegetative mass, and from a pH level of 5.5 up to strong basic (11) pH, the fungi developed colonies with an abundant vegetation mass [291]. It has been observed that spores of *Phomopsis viticola* are produced in a neutral (pH 7.0) to a strong basic pH (10.0–11.0). Spore-production was very good in the GCA medium, good on the malt extract, and absent on the Blakslee and Czapek culture media [304].

In regard to the temperature factor, Podosu et al. [304] observed a very good growth and spore production of the pathogen between 16 °C and 30 °C and placed the optimal growth temperature of *Phomopsis viticola* between 18 °C and 30 °C. Under low relative humidity conditions (31–38%), the growth of *Phomopsis viticola* was not substantial (0.1–0.4 cm in diameter), with white, feeble colonies and no spores’ production. At high levels of relative humidity (95%), the colonies became thick, normally developed, and with a good production of spores [304].

The energy sources (carbon source and light influence) of *Phomopsis viticola* growing in vitro are similar to those described for *Eutypa lata*. Regarding this matter, *Phomopsis viticola* showed best development and fructification on melibiosis, zaharosis, cellulose, and lignin growth mediums. Poor development and fructification of *Phomopsis viticola* was observed on sorbosis and maltosis growth medium. In the absence of the carbon sources, *Phomopsis viticola* colonies were inhibited [291]. The *Phomopsis viticola* colonies grown in completely dark conditions have developed the best vegetation mass, and the colonies showed dense, felt-like aspects, but they had weak fructifications. *Phomosis viticola* had very good fructification in alternative light and dark conditions (8/16 h) [291].

### 4.3. Esca in Romania

In Romania, Esca is treated as a vascular disease (the attack of the fungi blocks the xylem, diminishing at first and then completely blocking the circulation of water and nutrients through the vines), caused by several species of fungi, among the most common being *Stereum hirsutum* (Wild. [5,178,285,292,293]). They severely affect the wood of old and declining vines or those that have undergone periods of drought, extreme temperatures (heat, frost), a poor supply of nutrients, or those which have been planted on heavy soils with poor drainage [5,285,291]. Although *Stereum hirsutum* is not considered a vascular pathogen, the authors studying Esca in Romania report this fungus as being the main one causing the disease. *Stereum hirsutum* (Wild.) enters the vines through wood wounds caused by pruning or other accidents. Dead tissue is invaded by the fungus mycelium, which forms the macroconidia (the reproductive organs of the fungus) and sclerotia (the essential elements, the spores along with sclerotia, are released from the spongy mass, which is driven by the spread of the disease). The dry trunks crack, and the wind reaches wood wounds, infecting the tissues on the surface [178,285,293].

In Romania, this GTD manifests sporadically, in very old (20–30 years) vineyards, significant damage being rarely reported (Figure 7).

#### 4.3.1. Symptoms

External symptoms have very different aspects because the death of the vine can be slow, over several years, or sudden, by vine withering in a summer [5,285,293,307]. The chronic form, which is manifested on the leaves, is called Grapevine Leaf Stripe Disease (GLSD), and the acute form is called apoplexy [5]. The chronic form is more difficult to diagnose. Grapevine Leaf Stripe Disease (GLSD) begins to manifest in the second half of June, appearing at first on the basal leaves of the vines, in the form of isolated spots, small, chlorotic (in white varieties), or reddish (in red varieties) and irregularly arranged between the main veins. The spots turn necrotic and show a yellow halo between the healthy and affected area. Necrosis also occurs on the edge of the leaves, the attack being more pronounced on the leaves at the base of the shoots. The vines get poorly fed by sap; from year to year, portions of the vine do not start the vegetation period in spring, and finally, the vine dies [5,178,285,293].

Apoplexy is the most severe form of the disease, usually occurring in the hot and dry summer months (July, August) after heavy rains. It manifests as a sudden wilting of the leaves and drying of the entire vines in a very short period of only a few days. The leaves begin to lose their turgidity and dry out from the edge to the base, falling prematurely, while the vine remains bare. The fruit clusters wither and turn brown and remain attached to the canes for a while. The disease is manifested in solitary vines, the attack in outbreaks being very rarely observed [5,178,285].

The symptoms appear every year, but they are more obvious in the dry and hot years [5,178,285]. It is often observed that the vegetation starts from the rootstock or from the basal eyes of the graft [285]. The first symptoms can be observed around the flowering phenophase [178]. In summer, during periods of drought and wind, on the basis of increased evapotranspiration, the shoots and leaves begin to wither, having a reddish-brown appearance [5]. Before entering the varaison phenophase, on grape berries can appear some brown or brunviolaceeous spots. Similar spots also appear on rachis and pedicels. Berries lose their turgidity and begin to raisin. Sometimes their mummification can occur, as well as a longitudinal cracking of infected grains, which are then attacked by insects and rot [5].

The cross section of an Esca infected vine wood has a white-yellowish spongy appearance, surrounded by deep necrotic lesions, which secrete a gummy exudate on cutting. The mycelium of the fungus (*Stereum hirsutum*) breaks down the wood leading it to rot. The attack progresses year after year, forming annual attack zones, and when it has encompassed the entire circumference of the trunk, apoplectic drying of the vine occurs [5,178,285].

#### 4.3.2. Fungi Involved

The favoring factors of the appearance and development of Esca infections are: drought, heat, frost, relative humidity of air (more than 25%), optimal temperatures of 20–30 °C, the type of vine training system (Guyot is the most favorable), as well as grape cultivar (very sensitive are Muscat Ottonel, Sauvignon, Cabernet Sauvignon) and vineyards age [285,293,307]. As mentioned, pedoclimatic conditions play an important role in disease evolution [285,291]. Comșa et al. [293] observed that due to heavy rains and extreme temperatures during summer, a more extensive attack of the Esca disease took place in 2010. Oprea and Podosu [291] identified *Stereum hirsutum* in the Blaj (VZ1), Prahova (VZ3), and Ostrov (VZ7) vineyards, in a proportion of 8%, in a soil showing a high concentration of calcium carbonicum and ferric chloride and strong erosions. The disease of the vine was identified by the specific leaves color and the pathogen by the carpophores’ appearance on the leaves [291]. Moreover, Oprea and Dumitru [285] state that a higher number of dried vines were identified in vineyards with compacted soils. Mature vineyards are more susceptible to Esca, probably due to the higher content of tannins [285]. Matei et al. [307] observed that the type of vine training system can be a favorable factor, and the mode of pruning can favor disease spread. In their study, Esca decline was found in only 2.4% of vines trained by Cazenave cordon, in 3.7% of vines trained by demi-high Guyot as opposed to 13.58% in vines trained by spur-pruned cordon.

Oprea and Dumitru [285] found Esca symptoms in Iași (VZ2) vineyards (Comarna and Uricani), and they associate Esca GTD to *Stereum hirsutum* (Wild.). Moreover, they observed Muscadet and Cabernet Sauvignon cultivars as strongly attacked by Esca. Matei et al. [307] studied fungal pathogens associated with Esca in a 15 years plantation from Bucharest (VZ8) of Fetească Regală cultivar grafted on Kobber 5 BB and isolated nine fungal species from the wood of the grapevines with Esca symptoms. The isolates were identified according to their morphological characteristics on potato dextrose agar. The fungi isolated from the wood were included in the genera: *Phaeoacremonium*, *Phaeomoniella*, *Phomopsis*, *Fomitiporia*, *Fusarium*, *Alternaria*, *Cladosporium*, *Aspergillus,* and *Botryosphaeria*. Among those, *Phaeoacremonium aleophilum* and *Phaeomoniella chlamydospora* occurred at the highest frequencies. Their study suggests that the infection involving the *Phaeoacremonium* and *Phaeomoniella* species predisposes the vines to wood rots caused by basidiomycete fungi such as *Fomitiporia punctata*.

Comșa et al. [293] also link *Stereum hirsutum* to Esca infected vines from the Viticultural Centre of Blaj (VZ1), and observed that Fetească regală, Italian Riesling, and Sauvignon Blanc cultivars are more susceptible to the attack of lignicole pathogens, and Muscat Ottonel proved to be a more tolerant cultivar. Another study of Comșa et al. [295] states that *Stereum hirsutum* caused Esca complex on the cultivars Codreanca in Blaj vineyard (VZ1) and Traminer from Ciumbrud vineyard (VZ1). *Diplodia seriata* was also identified as a pathogen of the Esca complex on the cultivars Sultanina from Cuvin (VZ5) and Victoria from Blaj (VZ1) vineyards. *Phaeoacraemonium aleophilum* was also found as a pathogen in the infected grapevines of the cultivars Traminer from Ciumbrud (VZ1), Sultanina from Cuvin (VZ5), and Aligote from Sarica Niculitel (VZ6) vineyards, and it is considered to cause Esca complex dieback.

Ulea [178] claims that both *Stereum hirsutum* (Wild.) and *Phellinus igniarius* (L. ex Fr.) Quel. are pathogenic fungi isolated from trunks showing typical symptoms of apoplexy, and even if both were isolated in the same samples, one or the other may be more widespread depending on the region.

## 5. Other GTDs Pathogens Identified in Romanian Vineyards

The two fungal trunk diseases which are generally associated with young vineyards decline, Petri disease and Blackfoot disease, are barely acknowledged in Romania. Petri disease was identified in the Cuvin-Miniș vineyard on the rootstock Kober 5BB, in the Blaj-Târnave vineyard (VZ1) on the rootstock SO4-4 and in Sarica Niculițel (VZ6) on Aligote cultivar. The pathogens found responsible were *Phaeomoniella chlamydospora* and *Cadophora luteo-olivacea* on the rootstocks SO4-4 and Kober 5BB [295]. Blackfoot disease is described by Ulea [178] and attributed to *Nectria destructans* pathogen, but there are no signaled records of its incidence in Romanian vineyards. *Diplodia seriata* was also identified on the rootstock Kober 5BB and Aligote grapevine cultivar from Sarica Niculițel [295].

Tomoiaga and Chedea [300] also highlight *Phomopsis viticola*, *Eutypa lata*, *Stereum hirsutum,* and *Phellinus igniarius* as the most widespread and damaging pathogens reported in vineyards from the center of Transylvania (VZ1) and as sporadically identified, species such as *Phaeomoniella chlamydospora*, *Phaeoacremonium aleophilum,* and *Botryosphaeria obtusa*.

## 6. GTD Management in Romania

The link between GTDs, grapevine decline, and yield losses is now well established. These losses are a growing problem for viticulturists worldwide, as climate change puts vineyards under severe stress [21]. Because the GTD pathogens are located inside the wood, combating their attack is challenging. To current knowledge it is practically impossible to eradicate the GTD pathogens responsible for grapevines decline; however, it is possible to reduce contamination risks by acting on the points of pathogen penetration and taking prophylactic and cultural measures.

The most common practices used for GTD control can be grouped into preventive or post-infection methods according to whether the targeted vines are or are not affected. Mondello et al. [21] report all the practices applied by grapevine growers and indicate that some of them are not validated by experimental data. Among the commonly used practices, they highlight the following: as preventive practices for GTDs control in vineyards, the innovative approaches to vine training are the Guyot-Poussard and the sap flux-respect pruning and the multiple trunk training; the innovative pruning strategies are the early vs. late pruning, the double pruning, and the wound protection approaches [21]. Moreover, present research has become increasingly focused with the identification of GTD tolerating grapevine cultivars [300]. The management of GTD-affected vineyards includes remedial surgery, affected cordon or spurs removal, trunk renewal, trunk surgery or “curettage”, re-grafting, and pruning of symptomatic canes. The innovative tentative post-infection practices are hydrogen peroxide trunk injections, *Trichoderma*-inoculated wood dowels inserted into grapevine trunk, and copper nails in trunk [13,21,32,300,308,309]. No curative control measures are available to control GTDs in nurseries. Incorporation of multiple control measures, such as cultural practices and sanitation, chemical and biological control, hot-water treatments, and other strategies (e.g., ozonation), have been shown as the best approach to improve the phytosanitary quality of planting material [1,21,40].

In Romania, the management strategies are more or less the same as the ones described in the rest of the world. Given the fact that in Romania the most widespread GTDs are Eutypa dieback, Phomopsis dieback, and Esca, the adopted measures correspond to these. Regarding this matter, a limited amount of scientific literature is available. Oprea and Podosu [291] highlight the importance of pedoclimatic conditions, soil fertilization and structure, the vineyard vicinity, the vine pruning system, and the phytosanitary state of the vines and suggest the use of healthy viticultural material (from rootstock plantations and plantations supplying cane grafts free of pathogenic fungi) and special attention paid to adequate soil moisture, aeration, and fertilization as well as to the watering conditions.

In Romania, a high incidence of GTDs was recorded in areas where all of the above-mentioned factors were inappropriate. The inauspicious pedoclimatic conditions for the plant development proved to be a clay soil (in Blaj-VZ1, Jidvei-VZ1, Șard-VZ1, Miniș-VZ5, Valea Călugărească-VZ3, and Cotnari-VZ2), a weak soil fertilization with strong erosions (Diosig-VZ5 and Siria-VZ5), sandy soil structure favoring an insufficient watering condition, industrial pollution (Valea Călugărească-VZ3), prolonged drought (Murfatlar-VZ6 and Cotnari-VZ2 vineyards), and vicinity of the forest (Șard-VZ1 in Alba, Găiceanca-VZ2 in Bacău) [291].

### 6.1. Management of Eutypa Dieback

The Romanian National Phytosanitary Authority (ANF) [5] has specific management measures for each of the 3 main GTDs documented in Romania. Therefore, preventive measures for Eutypa dieback are the following:Prevention of stress conditions in the vineyard;Prevention of the infection and protection of the cutting wounds by wound sealing (painting or using an elastic sealing device);Avoid cutting in rainy weather when the release of fungal spores is at high levels;Renewal of infected vineyards (during the season, infected vines can be marked, and during winter they can be cut 20 cm above the grafting point; an offshoot that develops in the next growing season can be used to renew the vine; the wound in which the vine was cut must be covered with a sealing device);Removal and burning of the infected wood (dead wood must be cut during cutting season and then burned to reduce inoculation in the vineyard).

### 6.2. Management of Phomopsis Dieback

The management measures regarding Phomopsis dieback, from the perspective of the Romanian National Phytosanitary Authority (ANF) [5], include a part of preventive measures and one of curative measures. The preventive measures are:Setting up plantations and filling gaps with certified material, free of phytopathogens;Correct application of grapevine maintenance work (pruning, tying, hoeing, and weed control);Reduce the source of infection in the plantation (cutting and removal of the attacked vines and burning them);Ensuring an optimal plant density to allow good air circulation.

The curative measures include chemical methods. The following plant protection products are recommended: BISKAINE WG (2 kg/ha); ANTRACOL 70 WG (2.0 kg/ha); FOLLOW 80 WG (0.9 kg/ha to BBCH 61 and 1.8 kg/ha after BBCH 61); THIOVIT JET 80 WG (3.0 kg/ha). Good agricultural practices include the reduction of residues from agricultural products, thus break time from chemicals application to harvest is needed. In this regard, the Romanian ANF recommends (for the mentioned chemicals) the following break times: For BISKAINE WG—28 days; for ANTRACOL 70 WG—56 days; for FOLLOW 80 WG (0.9 kg/ha)—28 days; -THIOVIT JET 80 WG—28 days. [5]

Other curative measures for Phomopsis dieback were documented by Botea et al. [297] and Savu et al. [296]. Their studies were performed in Central Transylvania vineyards (VZ1) in Crăciunelu de Jos and Aiud-Ciumbrud vineyards.

The following cultivars were cultivated on Crăciunelu de Jos’s plots (0.5 ha): Sauvignon blanc, Italian Riesling, Fetească regală, and Traminer trained in demi-high Guyot system [297]. The treatments applied for Phomopsis dieback in Crăciunelu de Jos vineyard were caried out in the May–July 2020 period, and they are presented in Table 2.

The results revealed a pattern followed by all three cultivars: high attack degree (AD) at the start of the season, then a significant drop and a slow rise until it remained at a level that did not appear to affect the vines. The Traminer cultivar was an exception at the first evaluation, but the second and third evaluations followed the pattern described above. These results indicate that grapevine cultivars have different levels of tolerance to *Phomopsis viticola* attack, but the same responses to fungicide treatments. In this case Traminer proved to be more tolerant to *Phomopsis viticola* than Sauvignon blanc, Italian Riesling, and Fetească regală [297].

The second study, carried out in Aiud-Ciumbrud vineyards, aimed to evaluate the influence of the microclimate on *Phomopsis viticola* attack as well as its management with fungicide treatments. The treatments, presented in Table 3, were applied during the March–August 2020 period. The results conclude that *Phomopsis viticola attack* was present in Aiud-Ciumbrud vineyards, and it was influenced by microclimate conditions as well as fungicide treatments [296].

Tomoiagă and Chedea [300] conducted an important study regarding the GTD management in vineyards from Central Transylvania (VZ1). The main goal of this research was the development of new long-term management strategy for GTD control. To this purpose, an agroecological trial version was tested, and it included prophylactic and cultural interventions as well as two treatments with biofungicides based on *Trichoderma atroviride* strain 8, *Trichoderma atroviride* strain B11, and *Trichoderma harzianum*. The frequency of the main GTDs (Esca complex, Phomopsis dieback, Eutypa dieback) on the SCDVV Blaj homologated grapevine cultivars Astra, Amurg, Brumariu, Blasius, Radames, Rubin, and Selena was assessed, revealing that these had a lower frequency of symptoms than Fetească regală (the widely cultivated cultivar in this area). The impact of the training system (Classical vs. semi-high Guyot) on the main GTDs is also discussed in this study. The classical system reduced the incidence of Phomopsis dieback, while the semi-high Guyot system reduced the incidence of Eutypa dieback, demonstrating that the training system is an essential agroecological instrument in grapevine health maintenance. The biofungicides that have been tested seem to decrease the symptoms expression of GTDs, but they were not enough to prevent grapevine mortality. Overall, the study concludes that increasing the vine agro ecosystem’s resilience by using agro-ecological variants that include tolerant cultivars could be a viable option for reducing the biological decline of the vines caused by GTDs. Moreover, prophylactic preventive measures in combination with cultural measures such as the vine training system have a critical role in restricting the attack’s spread. According to their findings, applied after training, biopesticides based on *Trichoderma atoviridae* strain 8, *Trichoderma atoviridae* strain B11, and *Trichoderma harzianum* reduce the GTDs symptoms, but not enough to eliminate the issue of premature death of the vines. The biopesticide *Trichoderma harzianum* was efficient against Eutipa dieback, while *Trichoderma atroviride* strain B11 was effective against Phomopsis dieback [300].

### 6.3. Management of Esca Disease

In the matter of Esca GTD, the national phytosanitary authority (ANF) [5] states management measures of *Stereum hirsutum* (Wild.) Pers, as preventive measures, as follows:The use of healthy viticultural material (free of wood fungi);Limitation of the source of inoculum by removing and burning affected vines in windless periods in order to reduce the spread of fungal spores;Avoidance of excess nitrogen and lack of water;Spring pruning during periods without rainfall;Avoid strong wounds at spring pruning and treatment if they occur;Treatment of the vine after dry pruning to avoid infection with GTD pathogens;Reformation of the vine with the help of a healthy offshoot (only if the necrosis has not reached the rootstock). The success of the vine reformation is 75% for vines up to 25 years old.

## 7. Conclusions

At present, GTDs are high-profile diseases of the grapevine that seem to be increasing in incidence and severity worldwide. Taking into consideration that standard vineyards business models are generally based on 25 years, or more, of optimal productivity, GTDs are regarded as very important factors that limit the lifetime profitability expectancy of the vineyards, causing drops in productivity combined with a decrease in fruit quality and marketability. In Romania, even though this matter of grapevine decline was acknowledged since the beginning of the 19th century, the topic of fungal GTDs gained substantial awareness only in recent decades.

Worldwide there are acknowledged and well documented six fungal GTDs: Petri disease and Blackfoot disease which are associated with young vineyards (under 10 years) and Botryosphaeria dieback, Eutypa dieback, Phomopsis dieback, and Esca which are generally associated with mature vineyards (20–25 years). In Romania, only three of them are identified and documented in the scientific literature: Eutypa dieback (with the highest reported incidence), Phomopsis dieback (as the second reported incidence), and Esca (reported with sporadic manifestations). In terms of knowledge and understanding of Eutypa dieback and Phomopsis dieback, the situation in Romania is similar to the current state of knowledge from the rest of the world. With Esca disease, in particular, regarding the identification of the pathogens involved, there are some discrepancies. The authors studying Esca in Romania report *Stereum hirsutum* (Wild.) as being the causal agent of the disease, and although Esca is treated as a vascular disease, *Stereum hirsutum* is not a vascular pathogen. As far as we can discuss this issue, we can say that this situation is due to the fact that usually the symptomatic plants are old plants with white rot already set on the wood; *Phaeomoniella chlamydospora* and *Pheucremonium aleophilum* being slow growing fungi need special attention when performing wood isolations. Moreover, due to the reduced incidence of Esca disease, up to now, no Romanian research group focused on this subject.

All viticultural zones have reported Eutypa dieback, Phomopsis dieback, and Esca disease, except VZ4, for which there are no scientific data available up to now (Figure 8). Overall, the major fungal GTDs pathogens identified in Romanian vineyards include *Phaeomoniella chlamydosporum*, *Phaeoacremonium aleophilum*, *Campylocarpon*, *Cylindrocarpon destructans*, *Diplodia seriata*, *Eutypa lata*, *Phomopsis viticola*, *Fomitiporia mediterranea,* and *Stereum hirsutum*. Of these, *Eutypa lata*, *Phomopsis viticola,* and *Stereum hirsutum* are the most studied and well documented.

Symptoms are variable and often inconsistent, but they generally involve wood and leaf necrosis, as well as delayed development and/or poor growth, and in acute cases, grapevines death (Figure 3). Overall, the infection vectors and disease cycles of the pathogenic fungus outlined above are similar. The pathogens overwinter in diseased wood, and during high humidity periods develop fruiting bodies. Spores that are released by rainfall are then spread by rain splashes and wind. Moreover, it has been demonstrated that GTD pathogens can survive for larger periods of time in soils in a broad range of temperatures and humidity levels. Initial infection occurs on any fresh wound, and therefore, pruning wounds are considered the primary infection vector, especially due to the favorable climatic conditions during early spring pruning (high humidity, rainfalls). Wound size, weather conditions during pruning, grapevine age, and soil conditions and grapevine management system are all variables relevant to the risk of infection. The interaction between GTD pathogens and their environment is very complex and specific to cultivars and location, environmental stress having an important role with the potential of activating latent infections. The control measures are quite limited, and they mostly include preventive measures to stop the GTDs spread and the removal of affected grapevines (surgery of infected parts or removal of entire grapevines). Pruning wound treatments can minimize infection by restricting mycelial growth on open wounds, but the results are variable, depending on the local regulations regarding the use and availability of fungicides. Improper nursery practices, such as the use of infected mother blocks, are known to be a significant contributor to GTDs in young grapevines. Moreover, the cultural methods are an important part of viticultural good practices, for example, site selection of newly planted vineyards. Another viable approach for controlling the biological decline caused by GTDs is to increase the grapevine agro-ecosystem’s resilience by using agro-ecological variants that include autochthonous disease tolerant cultivars.

A large amount of capital and effort is required to establish a vineyard, and given GTDs conditions, it is entirely possible that the break-even point will be reached later than planned in the growers’ business model, or not at all, resulting in lower overall profitability. In order to avoid this problem and to support the growers in Romania, in 2010, we printed and also posted online on www.scvblaj.ro (accessed on 29 August 2022) a plant protection guide which helped the growers to gain knowledge about these diseases and the pests of the vine and that describes in detail the most important pathogens together with pictures and various solutions in order to help the growers to keep the plants healthy whenever possible.

## Figures and Tables

**Figure 1 pathogens-11-01006-f001:**
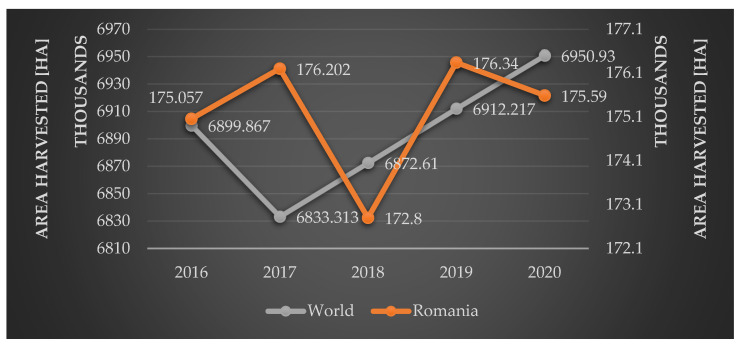
Areas cultivated with grapevines in the world and in Romania during 2016–2020.

**Figure 2 pathogens-11-01006-f002:**
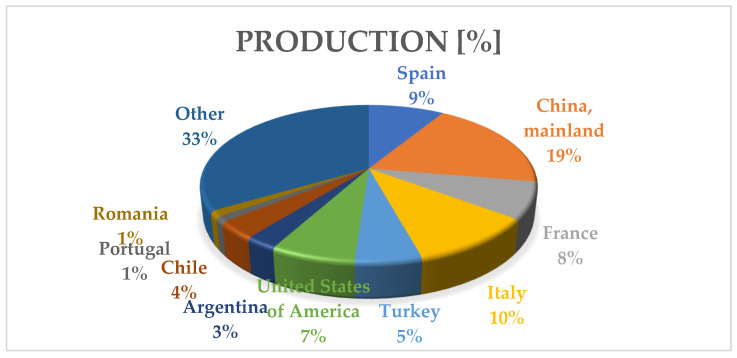
Ratios of grape cultivated areas held by the first ten ranked countries in the world in 2020.

**Figure 3 pathogens-11-01006-f003:**
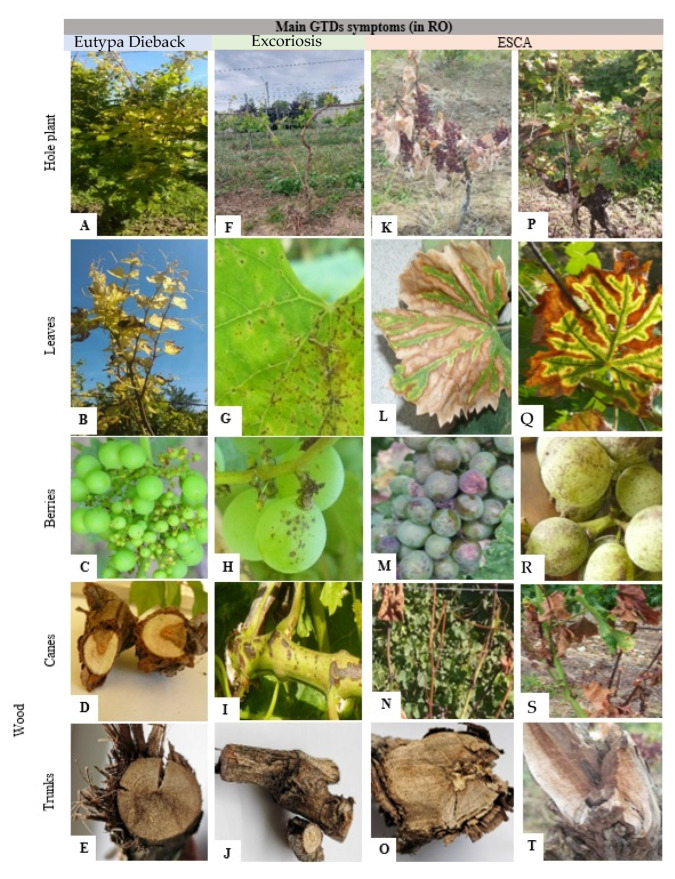
Illustrations of symptoms expression on different parts of the grapevines for the main GTDs observed in Romania: (**A**) Eutypa dieback symptoms on the entire vine; (**B**) Eutypa dieback symptoms on the shoots and leaves; (**C**) Eutypa dieback symptoms on berries/beaded berries; (**D**) Eutypa dieback symptoms on the canes; (**E**) Eutypa dieback symptoms on the trunk/cross-section of Eutypa dieback affected trunk; (**F**) Phomopsis dieback symptoms on entire vine; (**G**) Phomopsis dieback symptoms on leaves; (**H**) Phomopsis dieback symptoms on berries; (**I**) Phomopsis dieback symptoms on the canes; (**J**) Phomopsis dieback symptoms on multiannual wood; (**K**,**P**) Esca disease symptoms on the entire vine/K-acute form & P-chronic form; (**L**,**Q**) Esca disease symptoms on the leaves/L-white cultivar & R-red cultivar; (**M**,**R**) Esca disease symptoms on berries; (**N**,**S**) Esca disease symptoms on canes; (**O**,**T**) Esca disease symptoms on the trunk/(**O**) cross-section of Esca disease affected trunk and (**T**) Longitudinal section of Esca disease affected trunk.

**Figure 4 pathogens-11-01006-f004:**
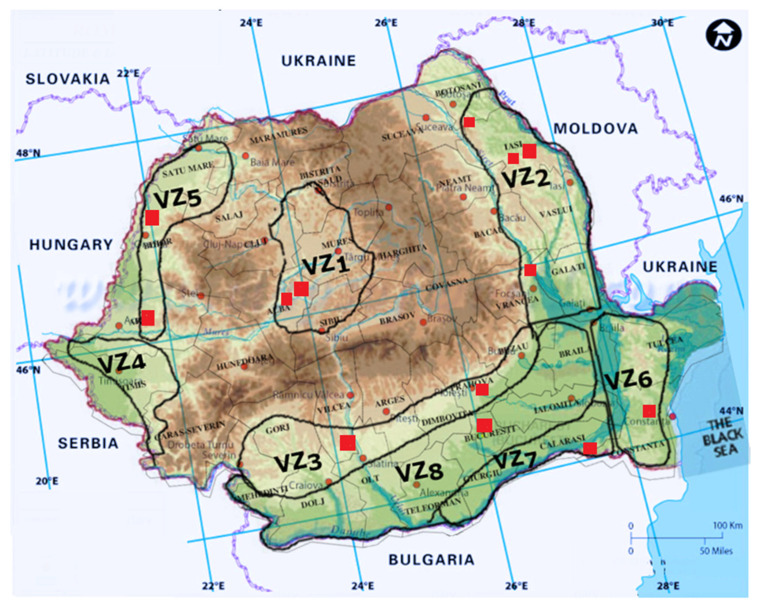
Graphical representation of Eutypa dieback reports in Romania.

**Figure 5 pathogens-11-01006-f005:**
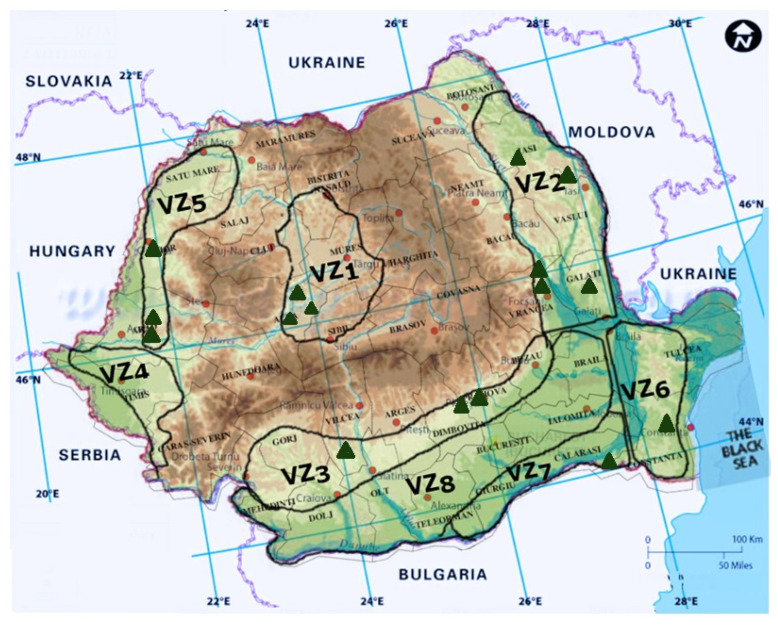
Graphical representation of Phomopsis dieback reports in Romania.

**Figure 6 pathogens-11-01006-f006:**
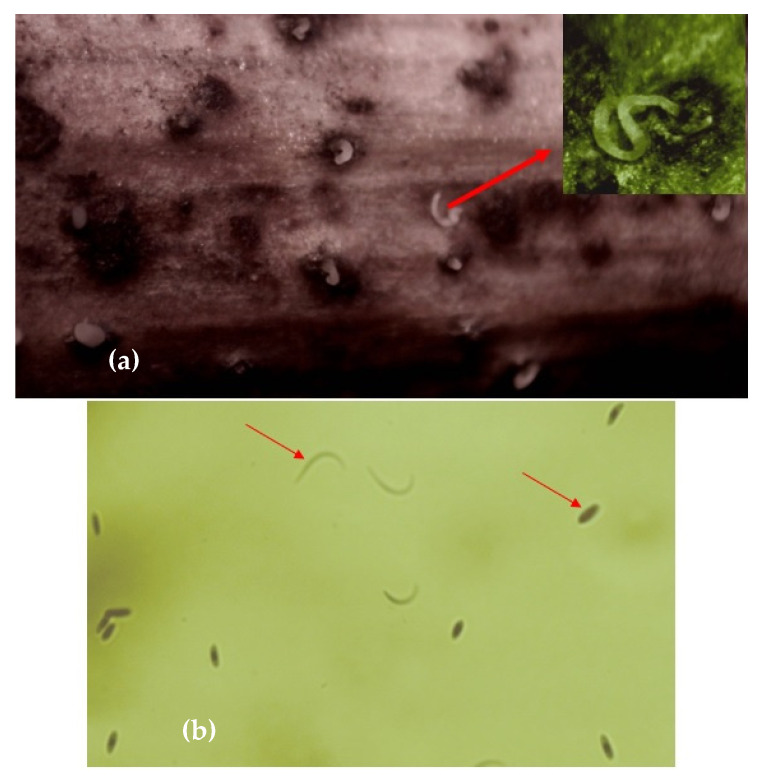
Pycnidia (**a**) and pycniospores (**b**) (α and β) of *Phomopsis viticola*.

**Figure 7 pathogens-11-01006-f007:**
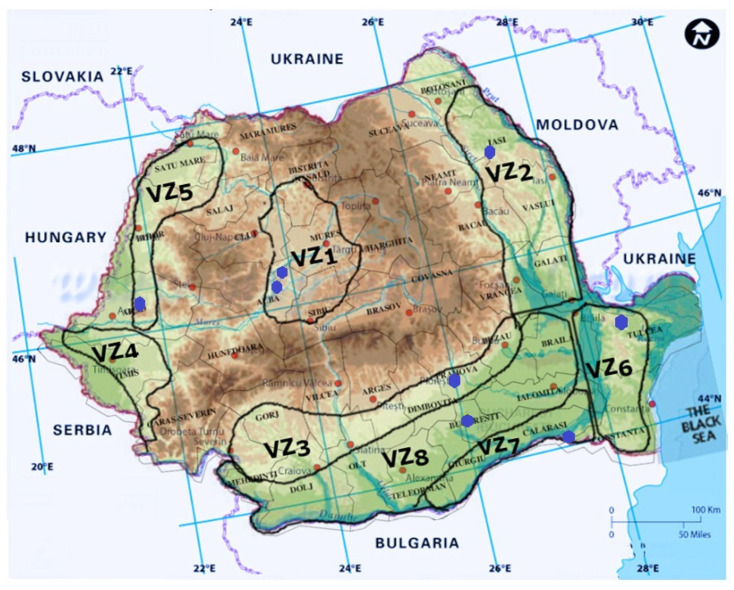
Graphical representation of Esca disease reports in Romania.

**Figure 8 pathogens-11-01006-f008:**
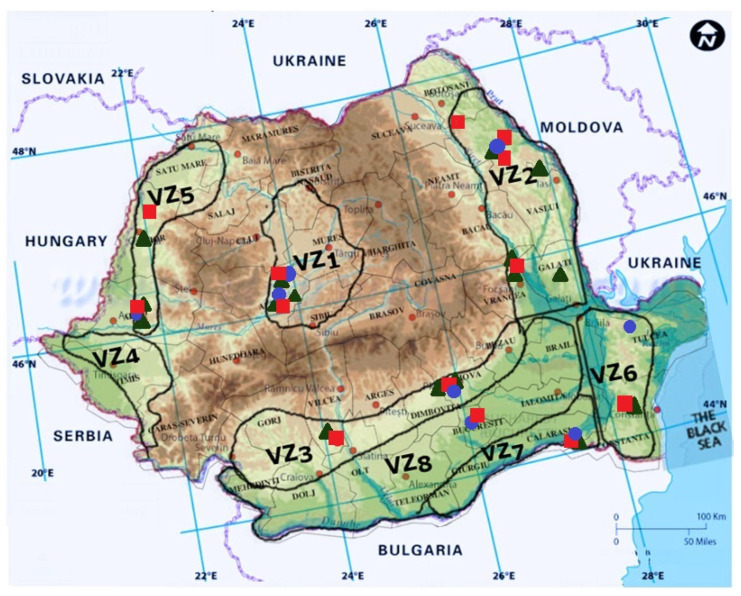
Graphical representation of main GTDs reported in Romania.

**Table 1 pathogens-11-01006-t001:** Viticultural zones (VZ) and vineyards in Romania [4,277,281].

Viticultural Zone	Harvested Area [ha]	Vineyards	Characteristics
*Transylvanian Plateau (VZ1)*	6800	Târnave	– suitable for obtaining quality and table white wines that are dry, semi-dry, and sweet as well as for sparkling white wines – recognized in the country and abroad, are the sparkling wines of Jidvei and Alba
		Alba	
		Sebeş-Apold	
		Ciumbrud-Aiud	
		Lechinţa	
*Moldavian Hills (VZ2)*	69,134	Cotnari	– the largest and most famous wine region in Romania – profiled on the production of wines and, to a lesser extent, table grapes. – the wines obtained, mostly white, are in a wide range, from current consumption to high quality naturally sweet – the Cotnari wine is included in the catalog of the best wines in the world. – dry wines are mainly made in the vineyards of Odobeşti, Coteşti, Panciu. – the production of red wines has an insular character.
		Iaşi	
		Huşi	
		Colinele Tutovei	
		Dealul Bujorului	
		Nicoreşti	
		Iveşti	
		Covurlui	
		Zeletin	
		Panciu	
		Odobeşti	
		Coteşti	
*Muntenia and Oltenia Hills (VZ3)*	53,450	Dealurile Buzăului	– Sâmbureşti vineyard, specializes mainly in the production of red wines – the other vineyards produce a wide range of mainly white wines (table and also quality ones)
		Dealu Mare	
		Ştefăneşti	
		Sâmbureşti	
		Drăgăşani	
		Dealurile Craiovei	
		Podgoria Severin	
		Plaiurile Drâncei	
*Banat Hills (VZ4)*	2930	Dealurile Banatului	– the smallest viticultural zone of Romania – counts as one vineyard – concentrated mainly on wine production, it is also suitable for table grape varieties – varieties worth mentioning are Chasselas, Muscat Hamburg, and Muscat of Adda
*Crişana and Maramureş Hills (VZ5)*	9100	Miniş-Măderat	– eco-climatic conditions allow the production of a wide range of wines, from the white and red ones of current consumption to those of superior quality. – suitable for the production of red wines is the south of the region, especially Miniş. – in Şimleul Silvaniei vineyard and partially in Zalău and also Maderat vineyard, the production of sparkling wines was imposed
		Diosig	
		Valea lui Mihai	
		Podgoria Silvaniei	
*Dobrogea Hills (VZ6)*	17,564	Sarica-Niculiţel	– relatively small region with best natural environment and heliothermic climate – many of the wines have been in demand for export since antiquity for their superior quality – represented by a large assortment of wines (red and white) and many grape varieties (table grapes and raisins cultivars)
		Istria-Babadag	
		Murfatlar	
*Danube Terraces (VZ7)*	11,234	Ostrov	– mainly cultivated with table grapes – the only zone in Romania that produces seedless raisins – the wine production is varied, consisting mostly of table wines, among which predominant are the white ones
		Greaca	
*Sands and other suitable terrains from the South (VZ8)*	12,960	Podgoria Dacilor	– the grape variety cultivated here consists mainly of wine varieties (white and red, table and quality wines) and, to a lesser extent, of table grape varieties.
		Calafat	
		Sadova-Corabia	

**Table 2 pathogens-11-01006-t002:** Treatments applied for Phomopsis dieback in Crăciunelu de Jos vineyard in 2020 [297].

Crt. No.	Date	Commercial Name	Active Substance
1	14 May 2020	Fantic M 0.25% + Thiovit Jet 80 WG 0.3%	Benalaxyl-M 4% + Mancozeb 65% Wetable sulfur 80%
2	22 May 2020	Polyram DF 0.25% + Topas 100 ec 0.025%	Metiram 70% Penconazol 100 g/L
3	28 May 2020	Equation PRO 0.04% + Topas 100 ec 0.025%	Cimoxanil 30% + Fomaxadon 22.5% Penconazol 100 g/L
4	11 June 2020	Universalis 593.5 SC 0.2%	Azoxistrobin 93.5 g/L + Folpet 500 g/L
5	19 June 2020	Mikal Flasch 75 WG 0.3% + Flint Max 75 WG 0.016%	Trifloxistrobin 25% + Tebuconazol 50% Aluminum fosetil 50% + Folpet 25%
6	29 June 2020	Mikal Flasch 75 WG 0.3%	Aluminum fosetil% + Folpet 25%
7	9 July 2020	Valis M 0.2%	Mancozeb 60% + Valifenalat 6%
8	19 July 2020	Triumf 0.25%	Copper hydroxide 40%

**Table 3 pathogens-11-01006-t003:** Treatments applied for Phomopsis dieback in Aiud-Ciumbrud vineyards.

Treatment No.	Active Substance
	*13 March 2020-first Phomopsis viticola frequency evaluation*
1	Copper Sulphate Pentahidrate + Sulphur 80%
2	Copper hydroxide + 50% Metallic copper. Abamectin 18 g/L Sulphur 80%
3	55% Metiram, 5% Pyraclostrobin Sulphur 80%
4	Fosetyl-aluminum 50% + Folpet 25% Sulphur 80% Cypermethrin (100 g/L) 240 g/L Myclobutanil Cyclohexanone
5	Trifloxystrobin 250 g/kg + Tebuconazole 500 g/kg Fosetyl-aluminum + Fluopicolide
6	Fosetyl-aluminum 50% + Folpet 25% 240 g/L Myclobutanil Cyclohexanone Cypermethrin (100 g/L) Boron 15%-ethanolamine
	*27 June 2020-second Phomopsis viticola frequency evaluation*
7	65% Mancozeb + 4% Benalaxil M + Metrafenone 500 g/L + Sulphur 80%
8	5% Mandipropamid 40% Folpet Sulphur 80% 300 g/L Fluxapiroxad
9	5% Mandipropamid 40% Folpet Sulphur 80% Fluopyram 75 g/L + Spiroxamina 200 g/L 500 g/L Clofentezine + Copper hydroxide + 50% Metallic Copper
10	Sulphur 80% + Copper sulphate pentahydrate
	*16 August 2020-third Phomopsis viticola frequency evaluation*

## Data Availability

Not applicable.

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
