# Peer review of "Fungal Grapevine Trunk Diseases in Romanian Vineyards in the Context of the International Situation"

_pathogens, 2022, doi:10.3390/pathogens11091006_

Round 1

Reviewer 1 Report

dear authors, the review deals with the Grapevine Trunk Dieseases in Romania. The work is well done but it is a little bit longer.

For each paragraph, it should have parts with title such as "pathogen inolved in", "description of symptoms", "..." Example of the refernce Bertsch et al., 2013, Plant disease.

The symptoms of GTDs in Romania should be presented at the beginning of part 4.

Author Response

Thank you very much for all of your suggestions. We have inserted titles according to the example you provided in all the parts that it was suitable. Also, we moved the figure illustrating the symptoms expression for the main GTDs observed in Romania just before the 4.1 part (line 682).

Reviewer 2 Report

The manuscript by Muntean and coauthor is well written and organized. It adds information on the  status of GTDs in Romania. A minor criticism arose from the absence of any reference about the chemistry work carried out during this 20 years of investigation, in order to shead light on the role of secondary metabolites or  or enzyme on the GTDs symptoms development . Fir this reason, I suggest to add the following literature in the introducing where the authors talk about symptoms development.

Masi, M., Cimmino, A., Reveglia, P., Mugnai, L., Surico, G., & Evidente, A. (2018). Advances on fungal phytotoxins and their role in grapevine trunk diseases. Journal of Agricultural and Food Chemistry, 66(24), 5948-5958

Author Response

Thank you very much for the comment. The recommended reference was already cited in the article at the time of manuscript submission (reference number [7]). However, a new paragraph was added to the initial text following the reviewer`s comment between lines 76-89.

Reviewer 3 Report

This review can be considered as a good addition for the knowledge of GTDs, since so little is known about the subject in Romania. These diseases have been extensively researched in other grape growing regions, and therefore this can constitute important new knowledge. Nevertheless, I have some comments and concerns regarding the review. First of all the state of the art in GTDs at a global level has many outdated references that could be updated. I left some comments on the PDF file regarding this.

Regarding the review of GTD in Romania, I understand that the authors are presenting a review of what has been done in the matter of GTD research in Romania. Nevertheless, I believe that the point of a review that compiles the state of the art of GTDs worldwide and in Romania, should also be to discuss what is the state of GTDs in Romania in comparison to what was found in the rest of the world. Specially since the information regarding, Esca in Romania is so contradicting with the research conducted on the rest of the world. The comments I left regarding specially S. hirsutum I think are a good example of aspects that I believe should be discussed to achieve a richer document. Also, I think that a proper discussion of the problematic of GTD situation in Romania vs worldwide is paramount and absolutely essential to indeed fulfill the objectives that the authors state on the abstract and introduction.

I therefore strongly encourage the authors to re-submit the paper, after preforming a review.

Author Response

Thank you very much for your comments and suggestions. Please find the response in the attached document.
